# Dynamics of transposable element accumulation in the non-recombining regions of mating-type chromosomes in anther-smut fungi

Marine Duhamel [1,2] ✉, Michael E. Hood[3,4], Ricardo C. Rodríguez de la Vega[1,4] & Tatiana Giraud[1,4]

In the absence of recombination, the number of transposable elements (TEs) increases due to less efficient selection, but the dynamics of such TE accumulations are not well characterized. Leveraging a dataset of 21 independent events of recombination cessation of different ages in mating-type chromosomes of *Microbotryum* fungi, we show that TEs rapidly accumulated in regions lacking recombination, but that TE content reached a plateau at ca. 50% of occupied base pairs by 1.5 million years following recombination suppression. The same TE superfamilies have expanded in independently evolved non-recombining regions, in particular rolling-circle replication elements (*Helitrons*). Long-terminal repeat (LTR) retrotransposons of the *Copia* and *Ty3* superfamilies also expanded, through transposition bursts (distinguished from gene conversion based on LTR divergence), with both non-recombining regions and autosomes affected, suggesting that non-recombining regions constitute TE reservoirs. This study improves our knowledge of genome evolution by showing that TEs can accumulate through bursts, following non-linear decelerating dynamics.

Transposable elements (TEs) are autonomously replicating DNA sequences ubiquitous in eukaryotic genomes[1–3]. TEs have occasionally been recruited by their host genomes, e.g., as regulatory sequences, for centromere maintenance or other functions[4–8]. However, TE replication and insertion are generally deleterious to their host[9–11]. TE proliferation is therefore often controlled by the host genome, by inactivating TE copies through epigenetic mechanisms such as RNAi silencing[12] and DNA methylation[13–16].

Deleterious TE insertions can also be purged *a posteriori* by recombination between homologous chromosomes carrying different TE insertions. In the absence of recombination, the average number of TE insertions in the population is predicted to increase with time[17]. TEs tend to accumulate in regions with lower recombination rates[18,19], particularly in non-recombining regions such as in centromeres[20], pericentromeric regions[21–23], and the non-recombining regions of sex chromosomes in plants, algae and animals[24–27] and of mating-type chromosomes in fungi[28–31]. The TE copies accumulating in the non-recombining regions of sex- and mating-type chromosomes may form a reservoir for the rest of the genome, with TE copies transposing from these regions onto autosomes when epigenetic defenses are insufficient[32–34].

The accumulation of TEs has been extensively documented, but the dynamics of their proliferation in non-recombining regions, and whether their accumulation in non-recombining regions has

[1]Ecologie Systématique Evolution, IDEEV, CNRS, Université Paris-Saclay, AgroParisTech, Bâtiment 680, 12 route RD128, 91190 Gif-sur-Yvette, France. [2]Evolution der Pflanzen und Pilze, Ruhr-Universität Bochum, Universitätsstraße 150, 44780 Bochum, Germany. [3]Department of Biology, Amherst College, 01002-5000, Amherst, MA, USA. [4]These authors jointly supervised this work: Michael E. Hood, Ricardo C. Rodríguez de la Vega, Tatiana Giraud. ✉e-mail: marine.c.duhamel@gmail.com

genome-wide impacts as a source of TEs, are unknown[35]. The cessation of recombination leads to degeneration in a variety of ways, which can have non-linear, decelerating dynamics as suggested for gene losses on theoretical grounds[35,36] and shown for optimal codon usage based on genomic data[37]. Alternatively, other components of degeneration can have a linear dynamics of accumulation, as shown for changes in protein sequences[37]. As TEs are a major degeneration factor, it is important to assess whether TEs accumulate rapidly in young non-recombining regions, whether the accumulation slows down over time, for example because of the evolution of more efficient control mechanisms, whether the number of TE copies grows exponentially out of control, versus linearly or by occasional bursts of proliferation. The curve shape for TE accumulation could thus inform our understanding of how their proliferation is controlled in the absence of recombination.

Chromosomes carrying the genes for reproductive compatibility are particularly interesting and useful models to address the temporal dynamics of TE accumulation in non-recombining regions. Closely related species can have independently acquired sex chromosomes with non-recombining regions of different sizes and ages, which have often extended in a stepwise manner, providing independent events of recombination suppression[38–40]. Such stepwise extensions of recombination suppression produce evolutionary strata of different levels of genetic differentiation between sex chromosomes[38,39,41].

The accumulation of TEs has even been suggested to play a role in the extension of recombination cessation along sex chromosomes[18,42]. TE-induced loss-of-function mutations constitute recessive deleterious mutations in the recombining regions adjacent to the non-recombining zone, called pseudo-autosomal regions. The segregation of deleterious mutations near non-recombining regions can foster the expansion of recombination suppression[43,44]. TEs can themselves promote inversions at the margin of non-recombining regions[45,46]. Additionally, epigenetic modifications that silence TEs, such as DNA methylation and heterochromatin formation, can also in themselves block recombination[36,42,47–50].

Similarly to sex chromosomes, fungal mating-type chromosomes can display large non-recombining regions, with evolutionary strata of different ages as well as footprints of degeneration[31,37,51–55]. Fungal mating-type chromosomes are particularly useful models for studying sex-related chromosome evolution as they often display young non-recombining regions, with many independent events of recombination suppression, which allows studying early stages of degeneration with multiple independent data points[37,52,53,56–58]. Fungal defenses against TE activity include several forms of RNAi-mediated post-transcriptional inactivation and heterochromatic silencing[59], in particular DNA methylation[15,60] and targeted C-to-T mutation in repetitive sequences, such as repeat-induced point mutation (RIP), targeting specific di- or tri-nucleotides[61–63].

*Microbotryum* fungi form a complex of closely related plant-castrating species, mostly infecting Caryophyllaceae plants, and multiple species have independently evolved non-recombining mating-type chromosomes[31,64–66]. In *Microbotryum* fungi, mating only occurs between haploid cells carrying alternative alleles at two mating-type loci: the pheromone-receptor (*PR*) locus controlling gamete fusion, encompassing genes coding for a pheromone-receptor and pheromones, and the homeodomain (*HD*) locus controlling dikaryotic hyphal growth, encompassing two homeodomain genes[67]. Most *Microbotryum* fungi display only two mating types, called $a_1$ and $a_2$[68], which is due to multiple independent events of recombination suppression having linked the two mating-type loci across the *Microbotryum* genus[30,31,37,64,66] (Fig. 1); a few of the studied *Microbotryum* species still have unlinked mating-type loci, but often with linkage of each mating-type locus to its centromere[65] (Fig. 1). Furthermore, patterns of decreasing divergence between the alleles

on alternative mating-type chromosomes when farther from the mating-type loci in the ancestral gene order have revealed that recombination suppression extended in a stepwise manner beyond mating-type loci in several *Microbotryum* fungi, forming evolutionary strata of different ages[31,37,64,66] (Fig. 1 and Supplementary Data 1). In contrast, the recombining regions of the mating-type chromosomes, called pseudo-autosomal regions, are collinear and highly homozygous in *Microbotryum* fungi[31,64,66]. The non-recombining regions of the mating-type chromosomes have accumulated TEs compared to the pseudo-autosomal regions and the autosomes[31,53]. However, whether specific TE superfamilies have accumulated and what are the temporal dynamics of TE accumulation across these genomic compartments remain unresolved questions.

At the whole-genome scale, *Microbotryum lychnidis-dioicae* carries long-terminal repeat (LTR) retroelements from the *Copia* and *Ty3* (also known as *Gypsy*, but see Ref. 69) superfamilies[62,63,70] as well as, to a lesser extent, *Helitron* DNA transposons, *i.e.*, rolling-circle replicating TEs[71]. In *Microbotryum* genomes, a RIP-like activity targets TCG trinucleotides (CGA on the reverse complement strand) rather than dinucleotides as in ascomycetes[62,72,73].

In this study, we investigate the temporal dynamics of TE accumulation in the non-recombining regions of 15 *Microbotryum* species, leveraging the existence of 21 independent evolutionary strata of different ages[37] (Fig. 1). We show that TEs rapidly accumulate in young non-recombining regions but that their abundance reached a plateau by 1.5 million years (MY) following recombination suppression at ca. 50% occupied base pairs. *Helitron* DNA transposons repeatedly expanded in non-recombining regions despite being rare in species with recombining mating-type chromosomes. The most abundant TE superfamilies, i.e., *Copia* and *Ty3*, are also over-represented in non-recombining regions and accumulated through multiple bursts that impacted both the non-recombining regions of the mating-type chromosomes and the autosomes at the same time. This concurrent increase, together with the positive relationship of TE content with the age of recombination suppression, suggests that non-recombining regions constitute a reservoir of TEs that transpose to recombining regions. We do not detect preferential TE accumulation at the margin of non-recombining regions.

## Results
### Transposable elements in *Microbotryum* genomes
We identified transposable elements (TEs) in the $a_1$ and $a_2$ haploid genomes deriving from a single diploid individual of each of 15 *Microbotryum* species and in a genome of the red yeast *Rhodothorula babjavae*, an outgroup with the ancestral condition of the *PR* and *HD* mating-type loci on separate chromosomes, using previously published high-quality genome assemblies[30,31,37,64–66]. We detected TEs de novo, using the MicroTEp pipeline implementing LTRharvest and RepeatModeler 1.0.11, combining results from three other programs: RECON, RepeatScout and Tandem Repeats Finder. The TE detection was enriched by blasting the TE models in genomes. We performed TE annotation using the fungal Repbase database 23.05 (see more details in Methods).

We detected no significant difference in TE content between the two haploid genomes within species (Wilcoxon paired test, $V = 5284$, $p$ value $= 0.208$, $n = 308$, Supplementary Fig. 1). In the following analyses, we therefore calculated the mean from the $a_1$ and $a_2$ genomes for each species. *Microbotryum* genomes, with an average of 18.3% base pairs occupied by TEs, displayed a higher density of TEs than the outgroup *R. babjavae* (Supplementary Fig. 1). *Microbotryum intermedium*, with fully recombining mating-type chromosomes, had the lowest TE content (6.57%), closely followed by *M. v. lateriflora* (7.02%) with a very young non-recombining region (0.304 My[66]), while *M. violaceum paradoxa* with the largest and among the oldest non-recombining

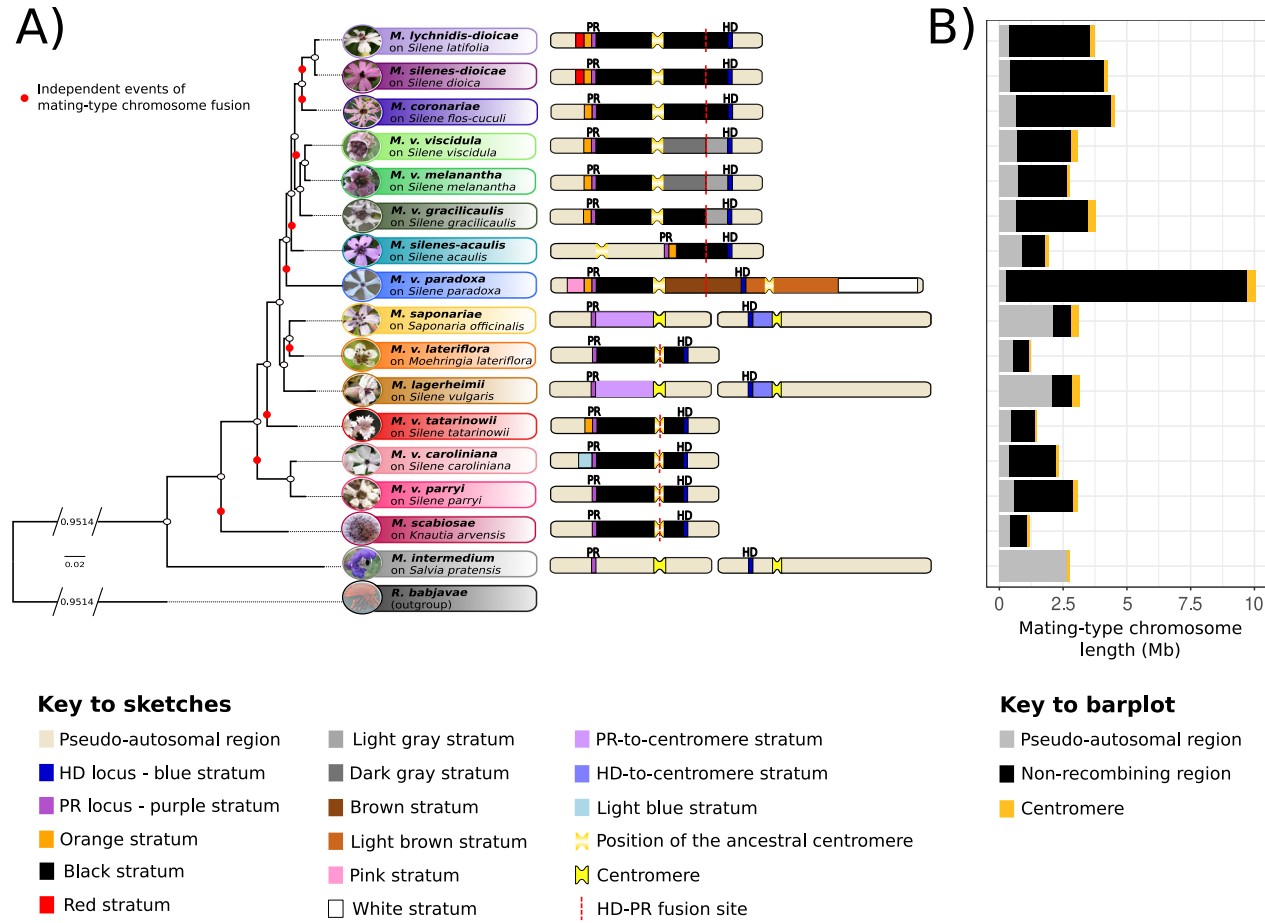

**Fig. 1 | Multiple evolutionary strata formed by independent events of mating-type locus linkage and stepwise extension of recombination suppression in *Microbotryum* fungi. A** The mating-type chromosomes of *Microbotryum* species with linked mating-type loci have had many rearrangements but are depicted here according to the gene order at the moment of the fusion between the homeodomain (*HD*) and pheromone-receptor (*PR*) chromosomes, the fusion site being indicated by a dashed red line. The ancestral gene order was inferred from *Microbotryum lagerheimii* and *M. intermedium* which have unlinked mating-type loci and collinear mating-type chromosomes. The old blue and purple strata have evolved before diversification of the *Microbotryum* clade. In *M. saponariae* and *M. lagerheimii*, recombination cessation linked each mating-type locus to their respective centromeres, forming the *HD*- and *PR*-to-centromere strata. The black strata correspond to the events of recombination suppression having linked the two mating-type loci, which occurred independently at least nine times in *Microbotryum*, indicated by red dots on the phylogeny. In some species (*M. violaceum viscidula, M. v. melanantha* and *M. v. gracilicaulis*), recombination between the mating-type loci was gradually suppressed following the chromosomal fusion event, and formed several strata called black, light gray and dark gray strata. In *M. v. paradoxa*, recombination was partially restored by introgression before being suppressed again, forming the brown stratum. In many species, recombination cessation further extended beyond the mating-type loci, capturing different sets of genes and forming strata that were named as indicated in the figure key. **B** Average ($a_1$-$a_2$) size of the mating-type chromosomes in *Microbotryum* fungi. The average ($a_1$-$a_2$) size of the pseudo-autosomal region, non-recombining region and centromere is represented in gray, black and yellow, respectively. Photos of the diseased flowers by M. E. Hood, except *M. v. viscidula* and *M. v. gracilicaulis* (by Hui Tang) and *M. v. parryi* (by acorn13 @iNaturalist, cropped). CC BY 4.0.

regions had the highest TE load (30.33%) (Fig. 2A and Supplementary Fig. 2 and Supplementary Data 2).

All *Microbotryum* genomes contained a substantial fraction of long-terminal repeat (LTR) retrotransposons from the *Copia* and *Ty3* superfamilies (6.04% and 4.97% of occupied base pairs on average for *Copia* and *Ty3* retroelements, respectively, Supplementary Data 2). These retroelements were particularly abundant in the *Microbotryum* genomes with non-recombining mating-type chromosomes, *Copia* TEs being for example ten times as abundant in *M. coronariae* (10.27% of occupied base pairs) as in *M. intermedium* (1.08% of occupied base pairs). Taken together, *Copia* and *Ty3* superfamilies represented 76.93% to 94.01% of the centromeric TEs across species (Fig. 2B). The trimodal distributions of *Copia* and *Ty3* retroelement length (Fig. 2C and Supplementary Fig. 3A) show that, while most copies are shorter than 1 kb and are likely degenerated and fragmented, a high proportion of these retroelements, being 5 kb and 7 kb length on average, are likely intact copies[74]. TE copies with a length greater than 10 kb likely represent nested TE copies and were particularly abundant in the young stratum of *M. v. paradoxa*, called the white stratum (Supplementary Fig. 3A).

Because retrotransposons such as LTRs can be common in plants[74,75], we investigated whether horizontal transfers of TEs could have occurred between the host plant and *Microbotryum* genomes. We compared the sequences from the plant Repbase database to the annotated TE sequences in *Microbotryum* genomes by blast. We obtained several BLAST hits with high identity and low e-value scores, but none exceeded 6% of the annotated TE sequence lengths (Supplementary Data 3). Therefore, no putative inter-kingdom horizontal TE transfer was identified.

DNA transposons were predominantly represented by *Helitrons* and terminal inverted repeats (TIR) elements in most *Microbotryum* genomes (1.44% and 0.42% of occupied base pairs on average, respectively), while these were nearly absent in *M. intermedium* (0.04% of base pairs occupied by *Helitrons* and no TIR elements) and in *M.*

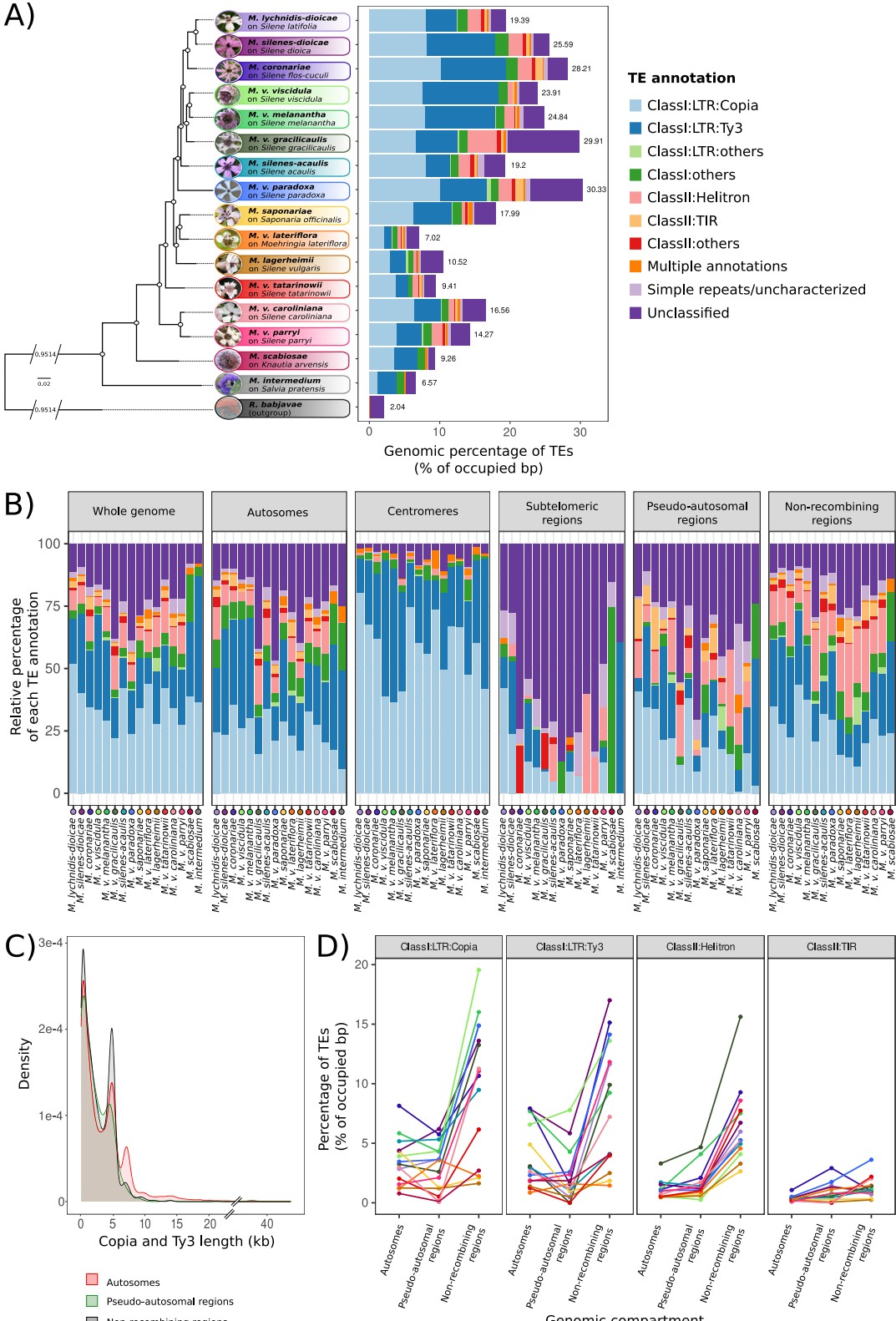

**Fig. 2 | Transposable element (TE) composition of *Microbotryum* genomes.**
**A** Whole-genome TE load (as percentage of occupied base pairs) across the
*Microbotryum* phylogeny, each color representing a TE category. Numeric values
correspond to the $a_1$-$a_2$ average. **B** Relative proportions of the various TE categories
in the different genomic compartments of *Microbotryum* genomes. Each color
represents a TE category, as in (**A**). **C** Distribution of the length of *Copia* and *Ty3*
retroelements in the autosomes, the pseudo-autosomal regions and the non-

recombining regions of *Microbotryum* fungi, in red, green and black, respectively.
**D** TE load per genomic compartment, *i.e.*, autosomes (without the non-
recombining centromeres), pseudo-autosomal regions and non-recombining
regions for the main TE categories (*Copia, Ty3, Helitron*, terminal inverted repeats –
TIR). The lines connect the values in the same species, to facilitate comparisons.
Each color represents a *Microbotryum* species.

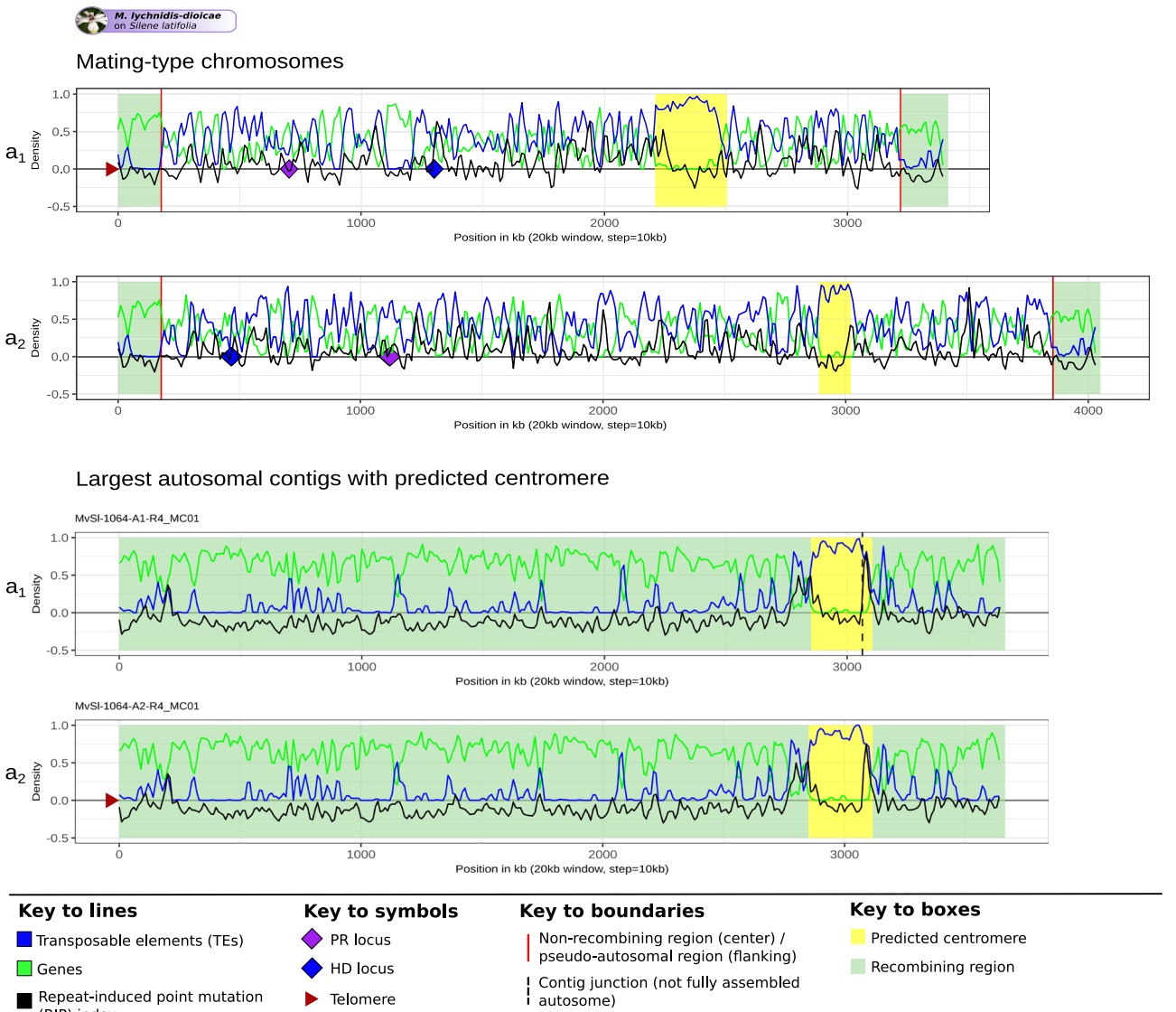

**Fig. 3 | Density of transposable elements (TEs), genes and repeat-induced point mutation (RIP) index along the mating-type chromosome and the autosomes of *Microbotryum lychnidis-dioicae*.** Density of TEs, genes and RIP-like index were calculated in 20 kb non-overlapping windows and plotted along the mating-type chromosomes and the largest autosomes, using the same X scale. A RIP-like index greater than zero indicates that the region is affected by a mechanism leading to C-to-T transitions resembling RIP. Predicted centromeres are indicated in yellow. Recombining regions of the mating-type chromosomes (pseudo-autosomal regions) and an autosome (the largest autosomal contig in the genome harboring a predicted centromere) are highlighted in pale green. The non-recombining region is delimited by red lines. Pheromone-receptor (*PR*) and homeodomain (*HD*) loci are indicated by purple and blue diamonds, respectively. Identified telomeres are indicated by brown triangles. The plots for the other species are shown in Supplementary Fig. 5.

*scabiosae* (Fig. 2A). The proportion of TEs annotated as "unclassified" was generally low (0.83% to 3.25% of occupied base pairs), being highest in genomes with high TE content, such as *M. v. paradoxa* and *M. v. gracilicaulis* (7.4% and 10.18% of occupied base pairs, respectively). The low average size of "unclassified" TE copies (263.13 bp, Supplementary Fig. 3B) suggests that they represent highly fragmented TE relics, being predominant in subtelomeric regions (Fig. 2B).

**Higher content of TE and RIP footprints in non-recombining regions**

Across the *Microbotryum* genus, TEs represented significantly higher percentages of base pairs in non-recombining regions compared to pseudo-autosomal regions and autosomes (ANOVA post-hoc Tukey test, adj. *p* value < 0.01, Supplementary Table 1), being on average 41.68% and 40.31% higher in the non-recombining regions compared to pseudo-autosomal regions and autosomes, respectively (Supplementary Table 1, Supplementary Fig. 4). No significant difference was

observed between pseudo-autosomal regions and autosomes (ANOVA post-hoc Tukey test, adj. *p* value = 0.922, Supplementary Table 1), despite tendencies in some species for lower content of some elements in pseudo-autosomal regions (Fig. 2D and Supplementary Fig. 4). The centromeres of all species displayed a high density of TEs (73% of base pairs occupied by TEs on average, Supplementary Data 4). Plotting the density of TEs along mating-type chromosomes further illustrated the higher values in the non-recombining regions than in the pseudo-autosomal regions, and that the TE-rich regions corresponded to gene-poor regions (Fig. 3 for *M. lychnidis-dioicae* and Supplementary Fig. 5A to O for the other *Microbotryum* species).

TE density was generally higher in older evolutionary strata such as the stratum around the *PR* locus, called the purple stratum (Supplementary Fig. 5H, J and O), and lower in young strata such as the pink and white strata of *M. v. paradoxa* (Supplementary Fig. 5G). In part because of this variation in TE content related to the non-recombining region age, the size of the mating-type chromosomes varied greatly

across *Microbotryum* species (Fig. 1B), from 1,177,022 bp in *M. scabiosae* to 10,060,889 bp in *M. v. paradoxa* (3,076,521 bp on average); the size of the mating-type chromosomes also varies because different chromosomal rearrangements were involved in the origin of *HD-PR* locus linkage[66]. For the same reasons, the size of the non-recombining regions was variable, ranging from 630,774 bp in *M. scabiosae* to 9,461,162 bp in *M. v. paradoxa*, thus spanning almost the full length of the mating-type chromosome in this later species (Fig. 1B), while being absent in *M. intermedium*.

Using base pairs occupied by TEs, we found no evidence that TEs have particularly accumulated in a 100 kb window of the non-recombining regions nearest the boundary with the pseudo-autosomal regions when compared to a window of the same size in fully recombining autosomes (Supplementary Fig. 6). Indeed, the probability of having a greater TE load in a 100 kb window on autosomes than at the margin of the non-recombining region was ≤ 0.05 in a single species (Supplementary Fig. 6). We compared the TE content in the flanking region of the non-recombining region to that in autosomes because the pseudo-autosomal regions were too small for building a distribution.

The TEs had more RIP-like signatures in non-recombining regions than in autosomes and pseudo-autosomal regions in most species (Fig. 3 and Supplementary Fig. 5). TEs in the youngest strata, such as the pink and white strata of *M. v. paradoxa* and the young light blue stratum in *M. v. caroliniana*, were in contrast less affected by RIP than those in the older non-recombining regions (Supplementary Fig. 5G, L, respectively).

## Particular TE family expansions in non-recombining regions

Some TE families have particularly expanded in non-recombining regions. *Helitron* elements, for example, have expanded in non-recombining regions, while they are relatively rare in the recombining mating-type chromosomes of *M. intermedium* and in autosomes in all species (ANOVA post-hoc Tukey test, adj. *p* value < 0.01, Supplementary Table 1 and Fig. 2B, D). TEs of the *Copia* and *Ty3* superfamilies displayed higher percentages of base pairs in the non-recombining regions compared to autosomes and pseudo-autosomal regions in almost all species (ANOVA post-hoc Tukey test, adj. *p* value < 0.01, Fig. 2D and Supplementary Table 1). On average, the content of *Helitrons*, *Copia* and *Ty3* elements increased from 1.12% to 6.53%, 3.41% to 9.97% and 3.73% to 8.51%, respectively, in non-recombining regions compared to autosomes, while the content in other TEs increased from 0.73% to 1.74% on average in non-recombining regions across species. The proportion of *TIR* DNA transposons was higher in the non-recombining regions compared to pseudo-autosomal regions but not compared to autosomes (ANOVA post-hoc Tukey test, adj. *p* value < 0.01, Supplementary Table 1 and Fig. 2D).

The elements with particular expansion in non-recombining regions, i.e., *Copia*, *Ty3* and *Helitron* were less affected by RIP than the other TEs (ANOVA post-hoc Tukey test, adj. *p* vlaue <0.001, Supplementary Fig. 7A, Supplementary Table 2). The lower abundance of RIP footprints for these elements compared to other TEs was more marked in autosomes than in the non-recombining regions (significant interaction in the ANOVA between genomic compartment and TE class, Supplementary Figs. 5 and 7B, Supplementary Table 2).

## Genome-wide TE bursts following recombination suppression

We found a significant and strongly positive correlation between the proportions of base pairs occupied by TEs on autosomes and non-recombining regions (two-sided Pearson's correlation test, *r* = 0.67; *p* value = 0.006; *n* = 15; Supplementary Fig. 8), which is consistent with the reservoir hypothesis, i.e., that the TE copies accumulating in non-recombining regions have a genome-wide impact by transposing to autosomes. The correlation alone, however, cannot give the direction of the effect or even a causality relationship.

We inferred TE genealogies of the *Copia* and *Ty3* superfamilies independently, using the alignment of their 5′-LTR sequences. Because the genealogies rely on the identification of the 5′-LTR sequences of the retrotransposons, only a subset of the *Copia* and *Ty3* retroelements were used to reconstruct the genealogies (on average 12% and 9.65% of the annotated *Copia* and *Ty3* superfamily retroelements, respectively, Supplementary Fig. 9) and genealogies were built for the species having sufficiently large sets of retained *Copia* and *Ty3* sequences.

The *Copia* and *Ty3* genealogies revealed the existence of many copies clustering within large clades having very low divergence, suggesting multiple bursts of transposition of these retroelements that impacted all genomic compartments (Fig. 4 and additional species shown in Supplementary Figs. 10 and 11), including in the species with old non-recombining regions (Fig. 4B and Supplementary Figs. 10G, H and 11B). The *Copia* elements from two different strains of *M. lychnidis-dioicae*, which share the same evolutionary strata with similar levels of synonymous divergence (Supplementary Fig. 12A and B), were inter-mingled within bursts, indicating that these burst are representative of the species rather than being strain-specific (Supplementary Fig. 12D).

Clusters of similar TE copies could be explained by either recent bursts of transpositions (that we define here as a sudden accumulation of copies, resulting from either an increase in intrinsic TE activity or from less efficient purge of new insertions) or by gene conversion events among old copies. We tested these alternative hypotheses by considering that conversion events among old copies would generate multiple similar copies with an old age while recent bursts would generate multiple similar copies with a similar age of insertion as their time of divergence. The age of a copy can be estimated from the divergence between 5′-LTR and 3′-LTR sequences as they are identical at the time of insertion and then diverge gradually with time. Similarly, the 5′-LTR sequences of a parental copy and its progeny copy are identical at the time of transposition and then diverge gradually (Fig. 5A). Therefore, copies resulting from a burst of transposition events should have accumulated roughly the same number of substitutions between their 5′- and 3′-LTR sequences as between their 5′-LTR sequence and the 5′-LTR sequence of their parental copy, i.e., their most similar TE copy. We therefore expect that, in case of transposition bursts without conversion events, plotting the number of substitutions between the 5′-LTR sequences of the most similar copies against the number of substitutions between the 5′-LTR − 3′-LTR sequences within copies would result in a cloud of points around the first bisector, with some stochasticity. We estimated the variance around the first bisector expected without gene conversion by simulating a cloud of points under a Poisson distribution. The points outside the expected cloud, at the bottom right, likely represent conversion events (low number of substitutions between 5′-LTRs from different copies but elevated number of substitutions between LTRs within copies).

In all species, we found only a minority of points at the bottom right outside of the Poisson-simulated cloud of points, for the clusters of similar *Copia* or *Ty3* copies (Fig. 5B, C, Supplementary Fig. 13). This suggests that conversion events are rare and indicates that the clusters of similar copies correspond mostly to bursts of transposition, due to increased activity or less efficient selection against insertions, rather than gene conversion among old copies. For several species, a few points fell on the top left corner outside of the Poisson-simulated cloud of points, likely corresponding to copies for which the closest related copy could not be identified. The colors indicating the compartment of origin of the pairs of copies are scattered on the plots, showing that conversion is not more frequent in any genomic compartment (Fig. 5B, C, Supplementary Fig. 13).

The recent bursts of proliferation show that *Copia* and *Ty3* retrotransposons are active in *Microbotryum* genomes. In several species, the centromeric TE copies tended to cluster together (Fig. 4 and Supplementary Figs. 10 and 11). In a few species, such as *M. v. paradoxa*

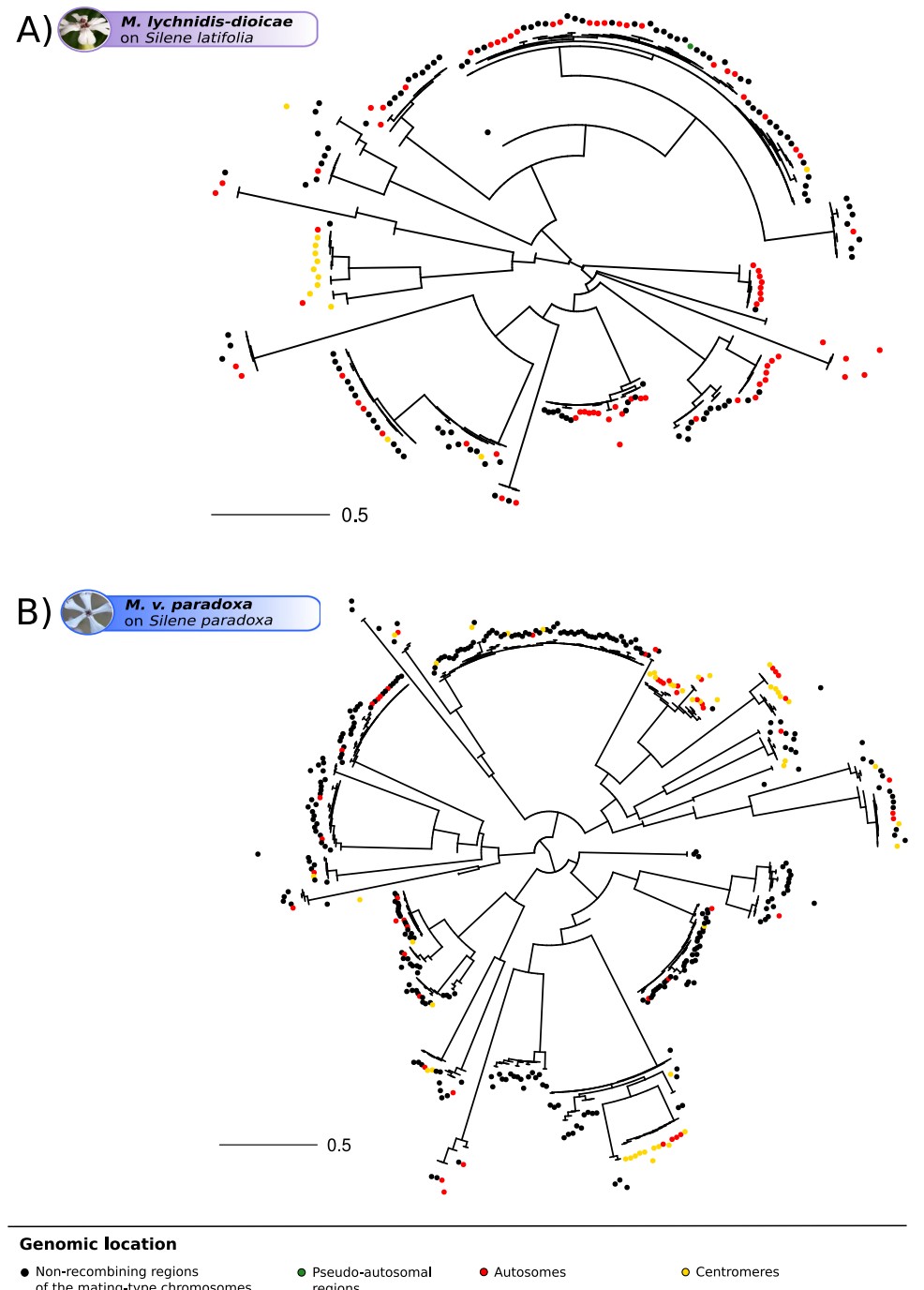

**A)** *M. lychnidis-dioicae* on *Silene latifolia*

0.5

**B)** *M. v. paradoxa* on *Silene paradoxa*

0.5

**Genomic location**

● Non-recombining regions of the mating-type chromosomes ● Pseudo-autosomal regions ● Autosomes ● Centromeres

**Fig. 4 | Single-species genealogies of *Copia* retroelement copies in *Microbotryum* genomes based on long tandem repeat (LTR) sequences.** Single-species genealogies of *Copia* copies in *Microbotryum lychnidis-dioicae* (**A**) and *Microbotryum violaceum paradoxa* (**B**). The color of the dots at the tip of the branches corresponds to the genomic location of the transposable element (TE) copies. Other single-species *Copia* and *Ty3* genealogies are shown in Supplementary Figs. 10 and 11, respectively.

(Fig. 4B), we identified *Copia* TE bursts specific to non-recombining regions. However, the clustering of copies from autosomes and non-recombining regions in the same TE clade with low divergence indicates that TEs transpose from one compartment to the other. We found similar distributions of pairwise divergence times between TE copies in autosomes and in the non-recombining regions of the mating-type chromosomes (Supplementary Fig. 14). This indicates that bursts simultaneously affected the two genomic compartments. We also found fewer pairs of most closely related copies with both copies in the non-recombining regions than expected by chance given the number of copies located in the non-recombining regions in the

genealogies (decreasing by 67% compared to random distribution on average, ranging from 95.62% to 40.6% across species, Supplementary Fig. 15). LTR divergence indicated that the TE copies were older in the non-recombining regions than in the autosomes (ANOVA post-hoc Tukey tests, adj. $p$ value = 0.01313 and adj. $p$ value = 0.00975, respectively, Supplementary Table 3), which is expected given that they are less often purged by selection in non-recombining regions and in agreement with the reservoir hypothesis.

In order to assess whether these TE bursts occurred following recombination suppression, we inferred TE genealogies for *Copia* and *Ty3* retroelements with the sequences of all species. The LTR

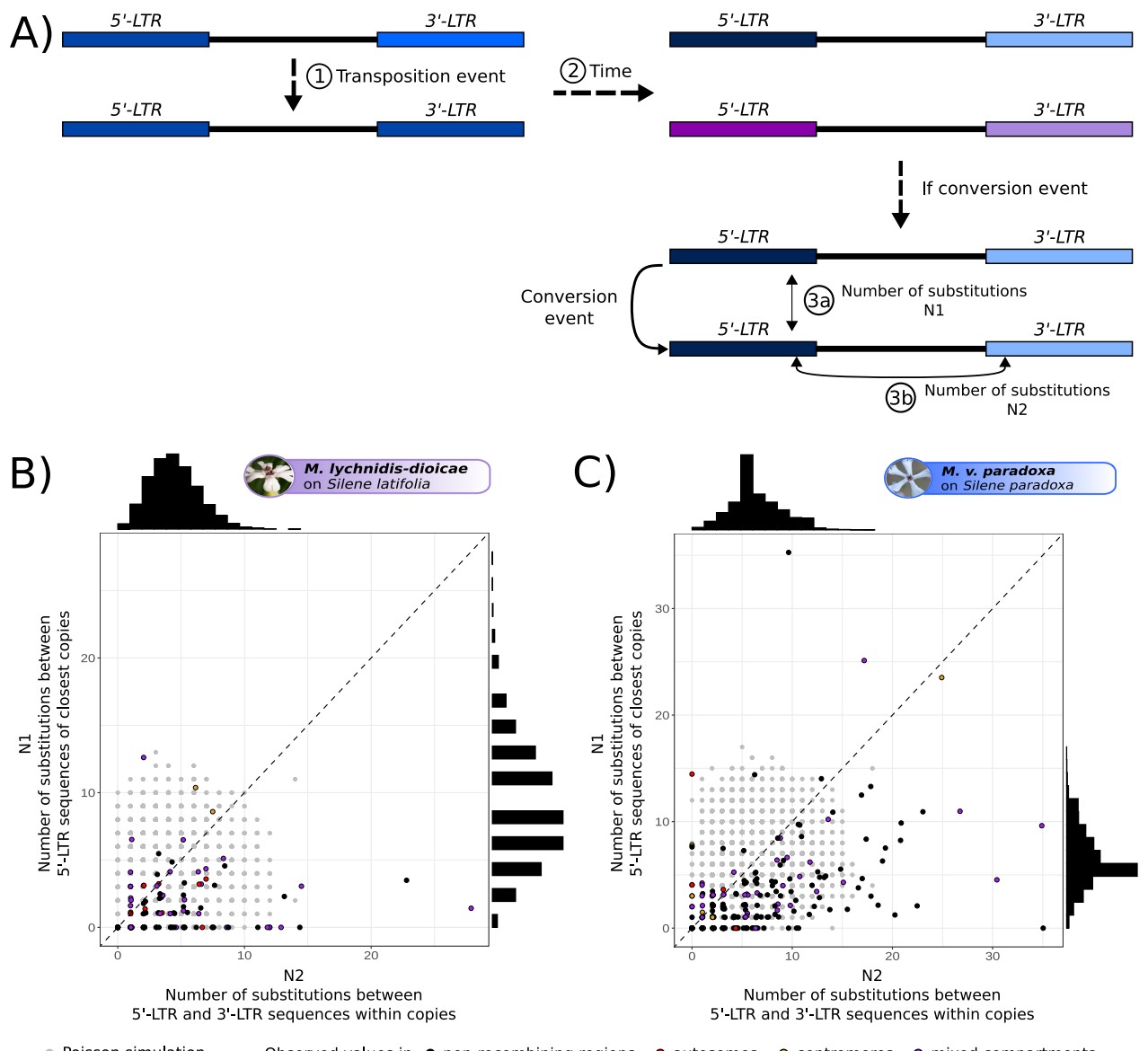

**Fig. 5 | Investigation of the possible occurrence of conversion events within clusters of highly similar LTR-retrotransposon copies. A** Illustration of LTR (long-terminal repeat) sequence divergence processes for retrotransposons: at the time of a new LTR-retrotransposon insertion, the 5′-LTR and 3′-LTR sequences are identical within the progeny copy and identical to the 5′-LTR of the parental copy (1). The LTR sequences then gradually diverge because they accumulate distinct mutations (2). The divergence between the 5′-LTR sequences of the two copies (N1) should thus be equal to the divergence between the 5′- and 3′-LTR sequences of the progeny copy (N2), the two measures being proportional to the age of the new copy. Following a conversion event, the 5′-LTR sequences between copies will be identical (3a) while the divergence between the 5′- and 3′-LTR sequences within copies remain high (3b). **B, C** Comparison of the observed number of different substitutions, between LTR sequences within (N2) and between copies (N1), in color, to the expected distribution under a Poisson model, in gray, with a lambda parameter equal to the average number of substitutions between 5′- and 3′-LTR sequences within copies, for the *Copia* retroelements in *Microbotryum lychnidis-dioicae* (**B**) and *M. violaceum paradoxa* (**C**). If no gene conversion occurred, we expect a cloud of points around the first bisector (y = x, dashed line) when plotting the number of substitutions between 5′-LTR sequences of the pairs of most similar copies within putative bursts against their 5′ − 3′-LTR sequence number of substitutions. In case of gene conversion, we expect points at the bottom right (low divergence between copies but old transposition ages). The points on the top left likely correspond to copies for which the closest related copy could not be identified. The colors of the points match the genomic location of the copy pairs. The observed distributions of the number of substitutions between 5′-LTR sequences of most similar copies and between 5′ − 3′ LTR sequences within copies are shown at the top and at the right, respectively.

sequences of the TE copies resulting from the recent bursts often clustered by species and by group of species deriving from the same recombination suppression event, supporting the inference that TE accumulation coincides with recombination suppression (Fig. 6 and Supplementary Fig. 16). The only exception was a clade with copies from *M. v. paradoxa* and *M. lychnidis-dioicae/M. silenes-dioicae* intermingled (Fig. 6, pointed by an arrow), which may be due to the introgressions previously detected in the *M. v. paradoxa* mating-type chromosomes[37]. The bursts of retrotransposon accumulation occurred at different times across lineages, as shown by the different phylogenetic distances of the clades with short branches to the root (Supplementary Fig. 17A, B). Three *Copia* bursts nevertheless occurred at the same time, in the three species of the *M. v. gracilicaulis* clade, as the three bursts appeared equidistant to the root (Supplementary Fig. 17A, pointed by arrows and aligned on the green circle).

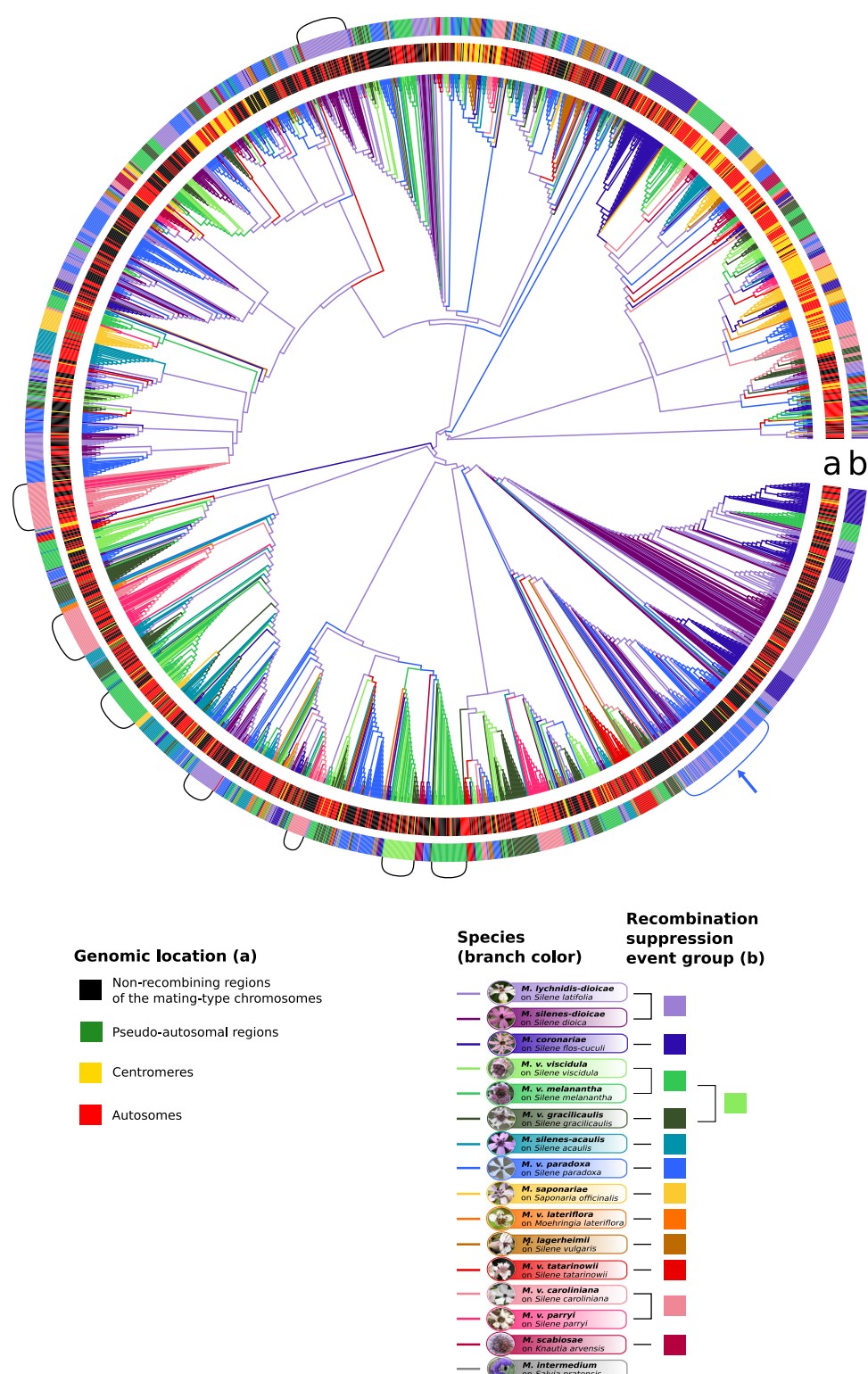

**Fig. 6 | Multi-species genealogies of *Copia* retroelement copies in *Microbotryum* genomes based on long tandem repeat (LTR) sequences.** Multi-species *Copia* tree of all *Microbotryum* species of this study. The branch color corresponds to the species, the first inner track (**a**) corresponds to the genomic location of the TE copies, the second outer track (**b**) corresponds to the linkage event group of the species carrying the TE copy. Brackets highlight bursts of TEs. The blue bracket and the blue arrow highlight the single burst with mixed copies from different recombination suppression events, with copies from *M. v. paradoxa* and the group formed by *M. lychnidis-dioicae* and *M. silenes-dioicae*. Multi-species *Ty3* tree of all *Microbotryum* species is shown in Supplementary Fig. 16. Multi-species *Copia* and *Ty3* genealogies with branch lengths are shown in Supplementary Fig. 17.

## Temporal dynamics of TE accumulation in non-recombining regions

The multiple independent events of recombination suppression across the *Microbotryum* phylogeny, with a range of ages, provides a unique opportunity to investigate the temporal dynamics of TE accumulation. The large non-recombining regions of different ages formed by the stepwise recombination suppression between the *HD* and *PR* loci were highly rearranged in *M. v. viscidula*, *M. v. melanantha*, *M. v. gracilicaulis* and *M. v paradoxa*, rendering difficult the assignment of some TE copies to a given age of recombination suppression. In order to be conservative, we only considered TE copies that could be assigned to a stratum with high confidence, by taking into account only blocks of at least 80 kb (size of the pink stratum, the smallest stratum of the dataset) and encompassing at least two genes of a given stratum. The resulting blocks represented 66.52% to 90.52% of the total non-recombining region sizes (Supplementary Fig. 18), thus likely yielding good estimates of their TE content. We excluded from analyses the orange stratum (Fig. 1) shared by all or most of the species[31,37,64–66], as it is small and likely not represent independent events of recombination suppression. We also excluded from analyses the small strata that were too fragmented (for example the red stratum in *M. lychnidis-dioicae* and *M. silenes-dioicae*; Fig. 1; Branco et al. 2018). To avoid pseudoreplication, i.e. for analyzing only independent data points, we considered the mean of the percentages of base pairs occupied by TEs in the $a_1$ and $a_2$ genomes for each species. Similarly, for the strata shared by multiple species, we took the mean values across the species derived from the same recombination suppression event. We used the TE content in the autosomes of *M. intermedium* as the point at time zero, i.e., before recombination suppression, as its mating-type chromosomes still recombine over almost their entire length, thereby precluding any potential TE reservoir effect.

We tested whether the percentage of base pairs occupied by TEs in the non-recombining regions could be explained by the time since recombination suppression and the ancestral size of the non-recombining region. We compared generalized additive models assuming linear or logarithmic relationships with the time since recombination suppression, with or without smoothing splines (i.e., phases), as well as a non-linear negative exponential model. The percentage of base pairs occupied by TEs in the non-recombining regions was best explained by the negative exponential model with the age of the stratum as an explanatory variable of the percentage of TEs in this stratum (Fig. 7 and Supplementary Table 4). We found that TEs rapidly accumulated in the first 1.5 million years following recombination suppression and then reached relatively abruptly a plateau at around 50% occupancy of TEs. A similar curve was obtained when fitting TE accumulation with the average synonymous divergence ($d_S$) between $a_1$ and $a_2$ associated alleles of the genes within strata, as expected considering the quasi-linear relationship between $d_S$ and stratum age (Supplementary Fig. 19). The ancestral size of the non-recombining region did not significantly impact TE content and we found no evidence for phylogenetic signal for TE accumulation among the independent evolutionary strata used in analyses ($p$ value = 1 when testing for $\lambda = 0$; $p$ value = 0.0011 when testing for $\lambda = 1$) in the retained negative exponential model.

We found that the relative abundance of RIP-like footprints (as assessed by the normalized RIP-index metric, see Methods) was highly variable in the young strata and remained constant with time since recombination suppression, with normalized RIP-index values greater than zero, suggesting that a process similar to RIP efficiently mutates TE copies in young non-recombining regions (Supplementary Fig. 20). There was indeed no significant correlation between the age since recombination suppression and the abundance of RIP-like footprints in the TEs of the strata (two-sided Pearson's correlation test, $p$ value = 0.9, $n$ = 22).

The fraction of *Helitron* copies that were intact decreased with time since recombination suppression and even reached zero after 4 MY (Supplementary Fig. 21). The proportion of copies of *Copia* and *Ty3* retroelements that were intact decreased rapidly following recombination suppression, from 20% down to 10%, and then remained stable at around 10% (Supplementary Fig. 21).

## Discussion

We found a large degree of variation among *Microbotryum* species in their overall TE load, and the key to understanding this variation seems to be the evolution of the recombination landscape in regions determining reproductive compatibility. Moreover, there appears to be a significant feedback of TE accumulation on non-recombining mating-type chromosomes where the buildup is rapid, and the rest of the genome is then burdened with greater repetitive element insertions. Leveraging a unique dataset with 21 regions having independently evolved recombination suppression, we found that TEs accumulated rapidly following recombination cessation but then reached a plateau at ca. 50% of occupied base pairs, after 1.5 MY.

We revealed more details on the degree of variation in genomic TE loads among *Microbotryum* genomes than previously recognized, ranging from 6.5% to 30% base pairs occupied by TEs, with an intermediate TE content in the most studied species, *M. lychnidis-dioicae* (19.4%). We were also able to annotate a large proportion of the detected TE copies, e.g. 89% in *M. lychnidis-dioicae*. In comparison, a previous study using Illumina paired-end sequencing estimated the genomic TE content of *M. lychnidis-dioicae* at 14.1% and only 60% of the TE copies were annotated[63]. Relative to other fungi, *Microbotryum* genomes carry intermediate TE contents, generally lower than their close relatives in the Pucciniomycotina[76]. We found that LTR retrotransposons, especially from the *Copia* and *Ty3* superfamilies, represented the majority of the base pairs occupied by TEs in *Microbotryum* genomes. These LTR retrotransposons summed to 87% of the base pairs occupied by TEs in *M. intermedium* and 70% in *M. lychnidis-dioicae*, which was much higher than previously estimated and among the highest LTR retrotransposon content in fungi relative to other TE categories[63].

Our results also indicate that the lack of recombination has allowed the accumulation of TEs by a lower efficacy of selection against deleterious insertions, as shown by the higher TE loads in non-recombining regions of mating-type chromosomes compared to recombining autosomal or pseudo-autosomal regions. The accumulation of TEs has been widespread in non-recombining regions of sex chromosomes in plants, algae and animals[24–27,77–79]. In the absence of recombination, the average number of TE copies can only increase with time in the population, because chromosomes devoid of new TE copies cannot be generated by recombination and the chromosomes carrying fewer TE copies can be randomly lost, a phenomenon known as Muller's ratchet[17,24,80,81]. Strong selection against TE copies disrupting essential gene function or rare back mutations may slow down Muller's ratchet but cannot stop it when population size is not very large. In *Drosophila miranda*, TEs played an early and important role in the degeneration of the neo-Y chromosome[25,82]. We detected RIP-like footprints in *Microbotryum* genomes, and at higher levels in non-recombining regions, but this genome defense mechanism was not sufficient to control TE proliferation in the absence of recombination, as previously noted in some *Microbotryum* species[70,72].

Transposable elements from the *Copia*, *Ty3* and *Helitron* superfamilies were particularly involved in the repeat accumulation in non-recombining regions. *Copia* and *Ty3* retrotransposons proliferated independently and repeatedly following the multiple independent events of recombination suppression across *Microbotryum* lineages. *Copia* and *Ty3* superfamilies are predominant in *M. intermedium* and in autosomes of the species with non-recombining mating-type chromosomes, suggesting that they were already abundant before

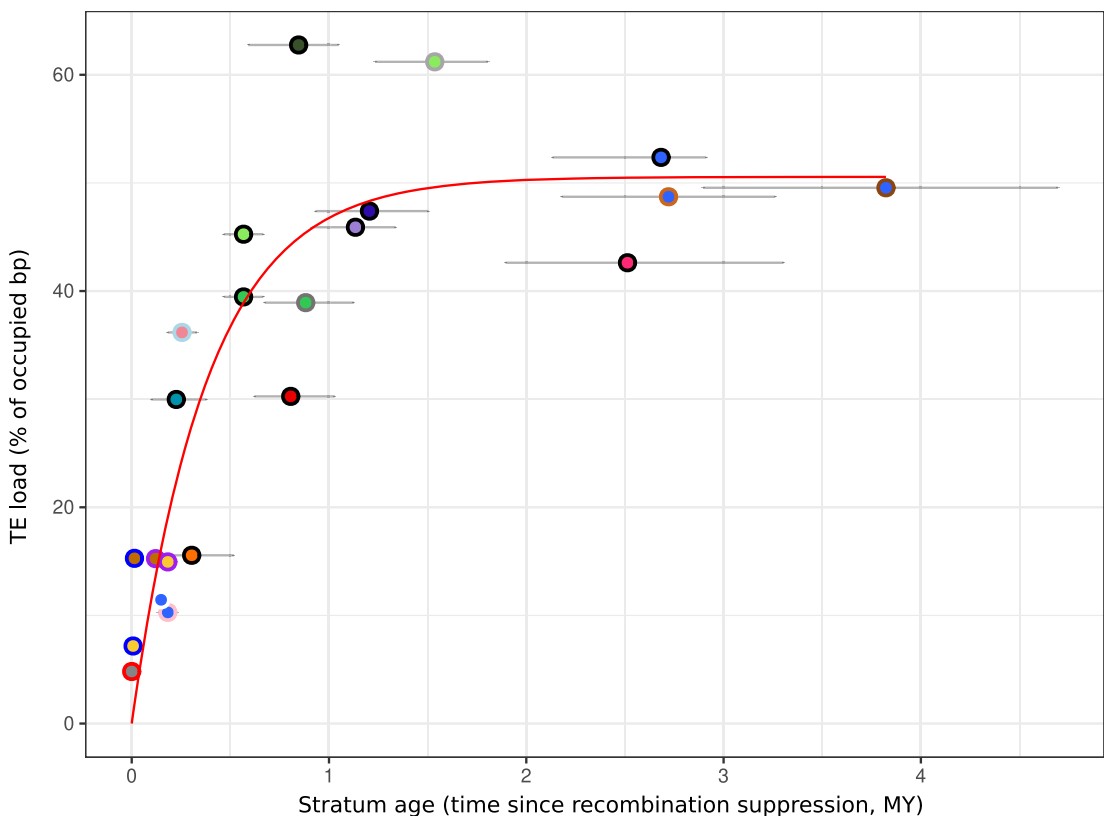

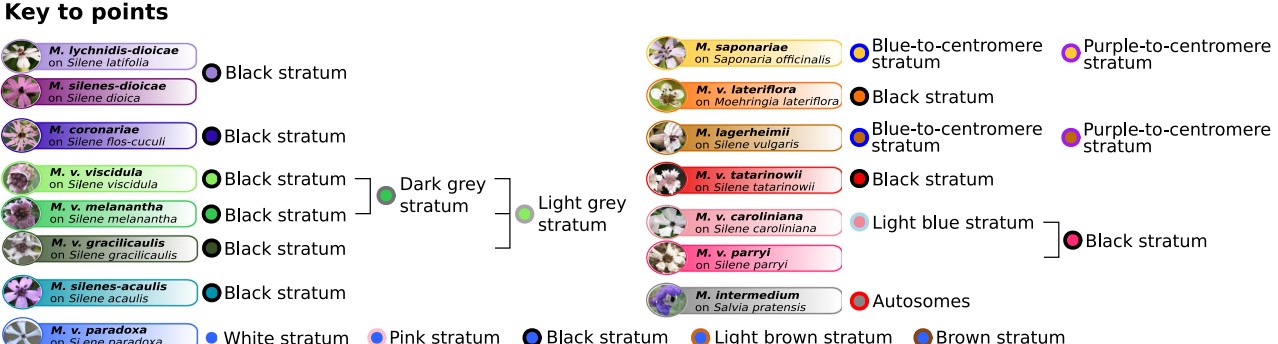

**Fig. 7 | Tempo of transposable element (TE) accumulation in the non-recombining regions of the *Microbotryum* mating-type chromosomes.** Percentage of base pairs occupied by TEs in non-recombining regions (means in $a_1$ and $a_2$ genomes and means across species sharing the same strata) as a function of their age in million years (MY) with confidence intervals. The red curve shows the prediction of the best model corresponding to a negative exponential model. Each dot corresponds to an independent evolutionary stratum. The error bars correspond to the 95% confidence interval of the age estimates of the evolutionary strata based on allele genealogies ($n = 10,000,000$ sampled trees).

recombination suppression, and then proliferated due to transposition and Muller's ratchet. These families had a lower abundance of RIP-like footprints, especially in autosomes. The particularly high accumulation of *Copia* and *Ty3* superfamilies in non-recombining regions may be because *Copia* retroelements preferentially insert in genes[83], so that they may be under particularly strong purifying selection in autosomes (and therefore all very young and hence not RIP affected), while selection is less efficient in non-recombining regions, or even relaxed due to sheltering by permanent heterozygosity.

*Helitron* elements were rare in *M. intermedium* and in autosomes of all species, and yet have also accumulated in the multiple non-recombining regions, as observed in the non-recombining regions of the sex chromosomes of *Drosophila pseudoobscura* and *D. affinis*[84]. *Helitrons* transpose by a mechanism similar to rolling-circle replication via a single-stranded DNA intermediate and have been suggested to preferentially insert in AT-rich regions; RIP and RIP-related

mechanisms increase the percentage in AT and are more abundant in non-recombining regions, which could have facilitated the accumulation of *Helitrons*[85]. It is striking that the same superfamilies preferentially expanded repeatedly in multiple non-recombining regions of *Microbotryum* fungi. Their specific mechanisms of transpositions may be particularly efficient in regions where recombination is suppressed, perhaps due to the chromatin landscape[83]. LTR retroelements for example preferentially target heterochromatin[86].

Through the inferred TE copy genealogies of the most abundant superfamilies in non-recombining regions (*Copia* and *Ty3* superfamilies), we found recent accumulation of TE copies through bursts that affected all genomic compartments, i.e., the non-recombining regions, the autosomes, and the centromeres, many of them being shared by species deriving from the same recombination suppression event. This suggests that TE accumulation in *Microbotryum* genomes is associated with recombination suppression, not only in the

non-recombining regions, but also in the rest of the genome. TE accumulation in non-recombining regions therefore likely has genome-wide impacts, as also shown by the strong significant positive correlation between TE load in the non-recombining regions and the autosomes. The direction of the effect is unlikely to be in the other direction, i.e., TE bursts in the autosomes that would invade non-recombining regions precisely when recombination suppression evolves, or this would imply that TE activity in autosomes positively correlated to the age of the non-recombining regions for a different reason than recombination suppression. In addition, we found that the TE copies were older on average in non-recombining regions than in autosomes. Our findings are thus, altogether, in agreement with the TE reservoir hypothesis[32,33], although direct evidence is still lacking.

Our TE genealogies were based on a fraction of the *Copia* and *Ty3* retrotransposon copies, due to the necessity to identify the 5'-LTR sequences. However, the 5'-LTR sequences were sometimes lost in degraded TE copies or could not be distinguished from the 3'-LTR sequence when the coding sequences were degenerated and could not be oriented. Nevertheless, considering the very low divergence between the TE copies within the genealogies, the burst patterns should not be affected by the sub-sampling of TE copies. Moreover, we found independent TE bursts across multiple *Microbotryum* species, indicating that the sampling of the TE copies used for the genealogies was good enough to capture accumulation events older than species divergence. Furthermore, the comparison of LTR sequence divergence within and between copies indicated that the clusters of similar copies were mostly due to recent bursts of transposition (due to increased TE activity or less efficient selection against their insertion), with only rare conversion events among copies.

We found similar TE contents in pseudo-autosomal regions compared to autosomes, without preferential accumulation of TE copies at the margin of the non-recombining regions. TEs have nevertheless accumulated everywhere in genomes following the formation of a non-recombining region. TE copies thereby accumulating at the margin of the non-recombining regions may impact their evolution[18,48–50]. TEs can trigger chromosomal rearrangements[87], and they recruit silencing marks, such as DNA methylation and heterochromatin formation, that can suppress recombination[18,36,42,47–50]. TEs can also induce deleterious mutations, which can select for recombination suppression that ensures the sheltering of the genetic load they create[43].

We found that TEs have rapidly accumulated in mating-type chromosomes following recombination suppression, but TE content reached a plateau relatively abruptly at about 50% of occupied base pairs after 1.5 MY. The rapid initial accumulation of TEs is expected: under the lack of recombination, new TE insertions cannot be purged, and each new copy can further make new copies[83,88]. The deceleration of TE accumulation after 1.5 MY can be due to several, nonexclusive phenomena: (i) selection against new insertions or for a more efficient control may be stronger when the load becomes higher, particularly if costs increase with copy number, or if TE insertions are more deleterious with time because genes become hemizygous; (ii) extant control mechanisms against TE proliferation can become more efficient with time by a cumulative effect, e.g. if methylation accumulates with generations[89] or could even evolve to become more efficient; (iii) new TE insertions nested in more ancient copies can further contribute to inactivate TEs, as suggested by the decreased proportion of *Helitrons* intact copies with time since recombination suppression in non-recombining regions; (iv) the chromatin landscape of non-recombining regions may become less favorable to new insertions, as for example histone modifications can affect the probability of new *Copia* retrotransposon insertions[83].

In conclusion, this study leverages a unique dataset of 21 independent evolutionary strata of different ages in closely related species to assess the dynamics of TE accumulation with time following

recombination suppression. We show that TEs have rapidly accumulated following recombination cessation and that the TE accumulation slowed down abruptly after 1.5 MY. We further show that some superfamilies repeatedly expanded in independent non-recombining regions, and in particular *Helitrons* that were not yet abundant before recombination suppression. *Copia* and *Ty3* superfamilies were also over-represented in non-recombining regions and have accumulated through bursts of proliferation, both in the non-recombining regions of the mating-type chromosomes and in the autosomes of *Microbotryum* species at the same time; this finding is in agreement with the TE reservoir hypothesis, although direct evidence is lacking. This study thus sheds light on important processes to improve our knowledge on genome evolution and in particular on the consequences of recombination suppression.

## Methods

### Detection and annotation of TEs

Transposable elements (TEs) were detected de novo in the $a_1$ and $a_2$ haploid genomes of 16 *Microbotryum* species, two strains of *M. lychnidis-dioicae* (1064 and 1318), and the red yeast *Rhodothorula babjavae* using LTRharvest[90] from GenomeTools 1.5.10, performing long-terminal repeat (LTR) retrotransposons detection and Repeat-Modeler 1.0.11[91] combining results from three other programs: RECON[92], RepeatScout[93] and Tandem Repeats Finder[94]. The TE detection was enriched by BLASTn 2.6.0+[95] using the genomes as a database and the previously detected TE models as queries. To fulfill the repeat criterion, a TE sequence detected by RepeatModeler or LTRHarvest and its BLAST hits were retained only if the query matched at least three sequences in the same species with an identity ≥ 0.8, a sequence length > 100 bp and a coverage ≥ 0.8 (defined as the query alignment length with removed gaps divided by the query length). When these criteria were met, we retained the other query matches for the following parameters: identity ≥ 0.8, sequence length > 100 bp, e-value ≤ 5.3e − 33 (25th percentile of e-value distribution) and coverage ≥ 0.8.

We performed TE annotation[2] using the fungal Repbase database 23.05[96] based on sequence similarity. We first performed a BLASTn search and a BLASTx search using the fungal Repbase DNA and protein sequence database, respectively, with TE sequences as queries. For each of these similarity-based searches, we set the minimum e-value score at 1e − 10 and minimum identity at 0.8. Then, we performed protein domain detection using pfam_scan.pl (ftp://ftp.ebi.ac.uk/pub/databases/Pfam/Tools/) on TE sequences and compared them to protein domain detection in the fungal Repbase database, keeping matches with e-values ≤ 1e − 5. We also considered the annotation found by RepeatModeler. In order to determine the best annotation of a TE sequence, we applied a majority rule: we chose the most frequent annotation, and, in case of equality, we kept both annotations, generating a multiple annotation; these multiple annotations can be due to TEs being inserted within other TE copies, generating nested TEs. When we found no annotation, we assigned the TE sequence to an "unclassified" category; we discarded such annotations when overlapping with predicted genes to avoid false positives. We also discarded predicted TE sequences annotated as rRNA or overlapping *Microbotryum* ribosomal sequences (downloaded from https://www.arb-silva.de/ and http://combio.pl/rrna/taxId/5272/) and mitochondrial DNA sequences (NC_020353) with e-value ≤ 1e-10. We used a python script for annotation, and the TE detection and annotation pipeline is available at https://gitlab.com/marine.c.duhamel/microtep.

### Phylogenomic species tree reconstruction

After removing the genes overlapping with annotated TEs, we built orthologous groups using Markov clustering (mcl 14-137)[97] of high-scoring pairs parsed with orthAgogue[98] based on all-vs-all BLASTP 2.6.0+[95] on protein sequences. We independently aligned the coding

sequences of 3,669 single-copy genes present in all *Microbotryum* species and the outgroup *Rhodothorula babjavae* using MUSCLE[99] as implemented in TranslatorX v1.1[100]. We used IQTREE 2.0.4[101] to build the maximum likelihood species tree with the GTR + GAMMA model of substitution chosen according to the Akaike information criterion by Model Finder implemented in IQ-TREE 2.0.4[102]. We used *Rhodothorula babjavae* as an outgroup. We assessed the robustness of the nodes with 1000 ultrafast bootstraps[103,104] and SH-like approximate likelihood ratio tests (SH-aLRT)[105] implemented in IQTREE 2.0.4[101] from the concatenated alignment.

### Evolutionary histories of *Copia* and *Ty3* retroelements

We inferred the evolutionary histories of the most abundant elements, i.e., in the *Copia* and *Ty3* retroelement superfamilies, by reconstructing their genealogies based on the sequence of their long terminal repeat (LTR) region at the 5′ end of the elements (5′-LTR), the rationale being that, at the time of transposition, the LTR sequences of the new copy are identical to the 5′-LTR sequence of its progenitor[106]. We only retained sequences harboring one or two LTR sequences identified by LTRHarvest[90], thus avoiding nested copies. We distinguished the 5′- from the 3′-LTR based on the orientation of their coding sequences detected using RepeatProteinMask[91]. Because some retrotransposon sequences were degenerated, their coding sequences could not be identified and thus oriented, so that only a subset of the retrotransposons were used for building the genealogies. We built multi-strain genealogies for the 1064 and 1318 strains of *M. lychnidis-dioicae* using the copies of *Copia* retrotransposons in the four haploid genomes (207 and 385 copies in total in each strain, respectively). We built multi-species genealogies using the TE copies of *Copia* and *Ty3* retrotransposons in all genomes (2468 and 694 copies, respectively). We built single-species (i.e., within genome) genealogies, only for species with at least 100 copies remaining after filtering (see proportion of TE copies used for the genealogies in Supplementary Fig. 9). We then aligned the 5′-LTR sequences of *Copia* or *Ty3* elements using MAFFT v7.310[107] (alignment length ranging from 816 to 2767 bp, 1678 bp on average). We used IQ-TREE 2.0.4[101] to infer the maximum likelihood tree from the 5′-LTR sequences alignment under the GAMMA + GTR model of substitution. We assessed tree robustness with 1000 replicates for ultrafast bootstrap and SH-aLRT. Trees were plotted using phytools 1.0-3[108] and ggtree 3.4.0[109]. Trees were displayed as rooted using midpoint rooting (*midpoint.root*, phytools 1.0-3[108]). Pairwise sequence divergence values in trees were obtained using *cophenetic.phylo* from ape 5.6-2[110].

We wanted to assess whether the identified clusters of TE copies with low divergence to each other originated from bursts of transposition or conversion events among copies. In case of gene conversion, multiple copies would be similar while their age of insertion would be ancient, if an old copy had copied itself by gene conversion into several other copies. In contrast, in case of recent bursts, all copies should have a young age of insertion in addition to be similar to each other. Because the 5′-LTR and 3′-LTR sequences are identical at the time of insertion and then diverge[111], we estimated the time since transposition of TE copies in the putative bursts by computing the number of substitutions between the 5′-LTR and 3′-LTR sequences, in the TE copies for which the 3′-LTR sequence could also be identified. For clusters of genetically similar copies, we compared the number of substitutions between 5′-LTR and 3′-LTR sequences per copy to the number of substitutions between the 5′-LTR sequences of pairs of most similar copies within the same cluster. For both measures, we computed the gamma-corrected Kimura 2-model parameter (γ-K2P), using MEGA11[112], and multiplied the divergence by the number of aligned sites. To compare the two values, we normalized the number of substitutions between the 5′- and 3′-LTR sequences by the number of aligned sites in the 5′-LTR sequences alignment. In case of bursts of transposition without conversion events, the 5′-LTR and 3′-LTR

sequences of each copy diverge gradually and at the same rate as the 5′-LTR sequences between copies from the same burst (Fig. 5A). Plotting the number of substitutions between 5′-LTR sequences of the most similar copies against the number of substitutions between the 5′-LTR − 3′-LTR sequences of each copy should therefore give a cloud of points around the first bisector. In case of conversion events, we expect that old copies (with divergent 5′-LTR and 3′-LTR sequences) have low divergence between one another while they have a high divergence between their 5′-LTR and 3′-LTR sequences, and therefore fall at the bottom right of the plot, far from the first bisector (Fig. 5A). To assess whether clusters of highly similar copies were caused by conversion events within clusters, we therefore plotted the number of substitutions between 5′-LTR sequences between closest copies as a function of the number of substitutions between their 5′- and 3′-LTR sequences (normalized by the same alignment length), for TE copies within clusters of genetically similar elements. As the number of substitutions in a sequence are rare discrete events following a Poisson law[113], we compared for each species this observed cloud of points to a cloud of 10,000 points generated by drawing their two coordinates from a Poisson distribution, with a lambda parameter equal to the observed average number of substitutions between the 5′- and 3′-LTR sequences within copies, as this value is not affected by potential conversion events among copies, but only by divergence since insertion. The points outside the simulated cloud, in the right bottom sector (low number of substitutions between copies but high number of substitutions within copies) would result from conversion events.

### Search for potential TE horizontal transfers

To search for potential horizontal transfers of transposable elements between *Microbotryum* fungi and their host plants, we compared the nucleotide sequences from the Repbase database 23.05[96] of Dicotyledones and *Arabidopsis thaliana* with the *Copia* and *Ty3* TE annotations from the *Microbotryum* genomes using BLASTN 2.6.0+[95].

### Centromere and telomere detection in *Microbotryum* genomes

We identified centromeres (Supplementary Data 5) as done previously[66] by blasting (BLASTN 2.6.0+[95]) the centromeric repeats previously described in *Microbotryum* fungi[30] on autosomal contigs larger than 20 kb; the largest stretch of centromeric repeats on a given contig was annotated as its predicted centromere. We delimited the predicted centromere by recursively extending the focal region by 1 kb intervals as long as gene density remained lower than 0.25 in the focal window. We detected telomeres (Supplementary Data 6) within 100 bp regions at the end of contigs carrying at least five times the telomere-specific TTAGGG motif (CCCTAA on reverse complementary strand), as done previously[30]. Subtelomeric regions were defined as the 20 kb regions adjacent to the telomeres, plus the 100 bp telomeres.

### RIP-like index

We computed values for a RIP-like index in non-overlapping 1 kb genomic regions along the mating-type chromosomes and autosomal contigs as previously described[114]. The RIP-like index was calculated as the ratio $t/n$, $t$ being the ratio of the RIP-affected sites (TTG + CAA trinucleotides, forward and reverse[62]) over the non RIP-affected sites (TCG + CGA, minus overlapping tetra-nucleotides TGCA) for the RIP target sites, and $n$ being the same ratio for non-RIP target sites ([A,C,G] TG + CA[C,G,T] over [A,C,G]CG + CG[C,G,T] - [A,C,G]CG[C,G,T]), to control for sequence composition. The RIP index was normalized as $t/n$ - *1*, so that a normalized RIP index greater than 0 was indicative of an excess of RIP-like mutations compared to random expectations. The script used to compute the RIP index is available at https://gitlab.com/marine.c.duhamel/ripmic. We used the *SlidingWindow* function of the evobiR 1.1 package (mean values, 10 kb steps[115]) and ggplot2 3.4.0[116] to plot the distribution of RIP along the mating-type chromosomes and autosomal contigs and identify the RIP-affected regions. We used the

same parameters to plot the density of genes and TEs in the same non-overlapping 1 kb windows. In order to test the relationship between the abundance of RIP footprint in TEs in non-recombining regions and the age of evolutionary strata, we calculated the average of the RIP index of the TEs in each stratum and performed a two-sided Pearson's correlation test.

## TE content in the flanking regions of non-recombining region

The distinct evolutionary strata within non-recombining regions have been previously distinguished based on the following criteria: i) the distribution of recombination suppression across the phylogenetic tree to assess when recombination suppression evolved and thereby identifying distinct evolutionary strata, i.e., genomic regions that stopped recombining at different times, and ii) the level of trans-specific polymorphism (i.e., alleles clustering by mating type rather than by species), which is also a strong indicator of the time of recombination suppression[31,37,64–66] (Supplementary Data 1). Young non-recombining regions, formed after the linkage of the *PR* and *HD* loci (i.e., the white and pink strata in *M. v. paradoxa* and the light blue stratum in *M. v. caroliniana*; Fig. 1) were not rearranged with other strata, so that it was straightforward to compute their percentages of base pairs occupied by TEs. In contrast, the ancient blue and purple strata (Fig. 1), having evolved at the base of the *Microbotryum* clade, were too much rearranged and intermingled with other strata for their specific TEs to be identified in species with large non-recombining regions, but they were very small (i.e., encompassing on average only 12 genes). We therefore pooled the ancient blue and purple strata with the larger strata with which they were intermingled. For the multiple independent evolutionary strata corresponding to the linkage of the *HD* locus with the *PR* locus in a single step (black strata; Fig. 1), it was straightforward to compute their percentage of base pairs occupied by TEs. In *M. v. viscidula*, *M. v. melanantha*, *M. v. gracilicaulis* and *M. v paradoxa*, recombination suppression extended gradually to link the *HD* and *PR* loci and then underwent multiple rearrangements that intermingled their genes and TEs. In order to avoid misassignment of TEs to these intermingled strata, we only considered regions spanning at least 80 kb (corresponding to the size of the smallest stratum from the dataset, the pink stratum in *M. v. paradoxa*) and with only genes from a single stratum. We defined the limit between two strata as the middle point between two genes from two distinct rearranged strata. The resulting blocks assigned to strata overall represented 66.52% to 90.26% of the total non-recombining regions formed by intermingled strata (Supplementary Fig. 18), which should thus give a good estimate of the stratum TE contents. We used circos 0.69-6[117] to visualize the blocks assigned to strata. For the non-recombining regions linking the *HD* or *PR* loci and their respective centromere (blue-to-centromere and purple-to-centromere respectively, in *M. lagerheimii* and *M. saponariae*), the percentage of base pairs occupied by TEs was calculated in the region between the end of the centromere and the mating-type locus. For each stratum, we took the mean percentage of TEs across $a_1$ and $a_2$ haploid genomes, and for the strata shared across several species, the mean of the $a_1$ and $a_2$ values across all the species sharing the same recombination suppression event. The ancestral size of the non-recombining regions were estimated by calculating the average size of the corresponding genomic region in the $a_1$ and $a_2$ haploid genomes of *M. intermedium*, i.e., the *Microbotryum* species most distant from the other species in our dataset, with unlinked mating-type loci and with little recombination suppression[66].

To test whether TEs also preferentially accumulate at the margin of the non-recombining regions, we compared for each species the fraction of base pairs occupied by TEs within the 100 kb of the pseudo-autosomal regions flanking the non-recombining regions to the distribution of TE load in 100 kb non-overlapping windows in fully recombining autosomes (100 kb away from the centromeres and after removal of 20 kb corresponding to subtelomeric regions). We only considered the species with a length of the pseudo-autosomal region at least equal to 100 kb (thus excluding *M. v. paradoxa*) and with at least 30 non-overlapping windows of 100 kb on their autosomes (thus excluding *M. scabiosae* and *M. v. parryi*, for which the genome assemblies were too much fragmented). We set the window length at 100 kb because this is the DNA fragment size at which linkage disequilibrium decreases below $r^2 = 0.2$ in *Microbotryum* populations[118]. The distribution of the 100 kb-window values on autosomes were plotted using the Sturges binning method.

We estimated the fraction of intact *Copia*, *Ty3* and *Helitron* copies in each stratum based on their length. For each stratum, we calculated the ratio of putative intact TE copies (5–7.5 kb for *Copia* and *Ty3* and 5–11 kb for *Helitron*) over the total number of *Copia* and *Ty3* copies or *Helitron* copies. For each stratum, we also calculated the mean fraction of intact TE copies across $a_1$ and $a_2$ haploid genomes and for the strata shared by several species, the mean across species deriving from the same recombination suppression event.

## Estimates of non-recombining region ages

In *Microbotryum* fungi, sexual reproduction occurs before the infection of a new plant, which will produce spores in its flowers at the next flowering seasons, so that one year corresponds to one generation for these fungi. We used the dates of recombination suppression estimated in previous studies[37,66], for 21 genomic regions corresponding to independent events of recombination suppression distributed across 15 *Microbotryum* species. The *M. scabiosae* non-recombining regions used in a previous study[37] was not considered here due to the large confidence interval on its estimated age. To this dataset of 15 *Microbotryum* species, we added the TE content in the autosomes of *M. intermedium* as the zero time point; indeed, the autosomes are recombining and there is little potential for a reservoir effect in *M. intermedium* as the non-recombining regions are very small, being restricted to close proximity around the separate mating-type loci[66]; we set the age since recombination suppression at 0.00000003 MY for *M. intermedium* autosomes, corresponding to one day since the recombination suppression event, to avoid the zero value for logarithm computation in temporal dynamics analyses.

## $d_S$ plot calculation

We calculated the synonymous divergence ($d_S$) from the alignment of $a_1$ and $a_2$ allele sequences using MUSCLE[99] implemented in TranslatorX v1.1[100]. We computed the $d_S$ and its standard error using the *ynOO* v4.9 f program of the PAML package[119] and plotted them using the ggplot2 2 3.4.0[116] in R. Standard errors were calculated per gene by PAML using the curvature method, i.e., by inverting the matrix of second derivatives of the log-likelihood.

## Statistical analyses

We tested whether the genomic percentage of base pairs occupied by TEs was different between the $a_1$ and $a_2$ mating-type haploid genomes using a paired two-sided Wilcoxon's rank test. To compare the percentages of base pairs occupied by TEs in the autosomes, the pseudo-autosomal regions and the non-recombining regions of the mating-type chromosomes, we performed a variance analysis (ANOVA) after testing the normality of the TE percentage distributions (Kolmogorov–Smirnov test). We then performed post-hoc Tukey's range tests to identify the pairs of genomic compartments which displayed significantly different TE loads, using the *TukeyHSD* R function providing adjusted p-values for multiple comparisons. We also performed variance analysis followed by post-hoc Tukey's range tests to compare the proportion of TE base pairs affected by the RIP in the autosomes, the pseudo-autosomal regions and the non-recombining regions, for the expanded TE categories versus other categories, and to compare the age of *Copia* and *Ty3* copies (number of substitutions between 5'-LTR and 3'-LTR sequences within copy) from the

genealogies between the non-recombining regions and the autosomes. We considered the species as a factor in both cases. We assessed the correlation between TE loads in autosomes and non-recombining regions using the two-sided Pearson's test.

In order to assess the shape of the functions for accumulation of TEs in the non-recombining regions of the *Microbotryum* mating-type chromosomes, we performed regressions using generalized additive models (*gam* function of R[120]) and non-linear regression models (*drm* function of the drc 3.2-0 package[121]). Based on the Akaike information criterion (AIC), we evaluated the model fits for predicting the percentage of base pairs occupied by TEs in the non-recombining regions as a function of the time since recombination suppression, with the ancestral size of the non-recombining region as a covariable, and assuming either a linear or a logarithmic relationship with the time since recombination suppression, with or without smoothing spline, or a negative exponential model. In order to test whether a phylogenetic signal could bias the analysis of the dynamics of TE accumulation with time, we performed a regression of TE content against stratum age using the *pgls* function of the caper package version 1.0.1[122] and an ultrametric tree of the evolutionary strata. We constructed the ultrametric tree based on the species tree topology[66] to which we added the splits corresponding to the following evolutionary strata: the black stratum shared by *M. v. caroliniana* and *M. v. parryi*, the two independent purple-to-centromere strata in *M. lagerheimii* and in *M. saponariae*, the brown, light-brown, black and pink strata in *M. v. paradoxa*, the light-gray stratum shared by *M. v. gracilicaulis, M. v. melanantha* and *M. v. viscidula*, and the dark-grey stratum shared by *M. v. melanantha* and *M. v. viscidula*. Branch lengths were taken from the corresponding ultrametric strata trees[37,66].

In order to assess the relationship between the fraction of intact *Copia*, *Ty3* and *Helitron* copies and time since recombination suppression, we performed a local regression using the *loess.as* function from the fANCOVA 0.6-1 package, using the Akaike Information Criterion for automated parameter selection[123,124].

We performed all R analyses in R version 4.2.1.

### Reporting summary

Further information on research design is available in the Nature Portfolio Reporting Summary linked to this article.

## Data availability

All data generated or analyzed during this study are included in this published article and the supplementary information files (Supplementary Tables 1–4 and Supplementary Data 1–6). Datasets in figshare are described in Supplementary Note 1. The rRNA or overlapping *Microbotryum* ribosomal sequences were downloaded from https://www.arb-silva.de/ and http://combio.pl/rrna/taxId/5272/ and mitochondrial DNA sequences from GenBank (NC_020353). TE sequence similarity searches were performed using Repbase database 23.05. Sequencing data and genome assemblies were published previously and available at GenBank under the BioProjects PRJEB12080, PRJNA437556, PRJEB16741, PRJEB15277, PRJNA771266 and PRJNA437556. All fungal strains are available upon request. Source data are provided with this paper.

## Code availability

The TE detection pipeline is available at https://gitlab.com/marine.c.duhamel/microtep. The script used to compute the RIP index is available at https://gitlab.com/marine.c.duhamel/ripmic.

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

## Acknowledgements

We thank Dominik Begerow, Sebastian Klenner, Lena Steins, and Frederick Witfeld for providing useful comments on the manuscript. We thank Gilles Fisher, Paul Jay, Jacqui Shykoff, and Olivier Gascuel for useful discussions. This work was supported by the National Institute of Health (NIH) grant R15GM119092 to M. E. H., the Louis D. Foundation award and the EvolSexChrom ERC advanced grant #832352 to T. G. and a public PhD grant to M.D. overseen by the French National research Agency (ANR) as a part of the "Investissements d'Avenir" through the program "Action Doctorale Internationale, ADI 2018" project funded by the IDEX Paris-Saclay (ANR-11-IDEX-0003-02).

## Author contributions

M.D., T.G., R.C.R.d.l.V. and M.E.H designed the study. T.G., R.C.R.d.l.V. and M.E.H supervised the study. M.D. performed the genomic and statistical analyses and produced the figures. R.C.R.d.l.V. advised in genomic analysis and data visualization. T.G. and M.D. wrote the original draft. All authors contributed to the manuscript.

## Competing interests

The authors declare no competing interests.

## Additional information

**Peer review information** : *Nature Communications* thanks Vincent Colot and the other, anonymous, reviewer(s) for their contribution to the peer review of this work. A peer review file is available.

