## [Peer Review file · Nature Communications]

REVIEWER COMMENTS

Reviewer #1 (Remarks to the Author):

This manuscript by Duhamel et al presents a comprehensive analysis of the dynamics of TE accumulation within non-recombining regions of the genome of 15 anther-smut fungal species. Specifically, the authors have exploited whole genome long-read sequencing data previously obtained for 15 *Microbotryum* species where regions of recombination suppression between mating-type loci have evolved independently at different times in the past to determine the potential impact of such recombination suppression on TE accumulation. As expected, they observe a very clear time-dependent overaccumulation of TEs within non-recombining regions compared to the rest of the genome, which is mostly contributed by Copia and Ty3 superfamilies of LTR retroelements and Helitron rolling-circle DNA transposons. More precisely, there appears to be a rapid increase in TE density within 1.5My of recombination suppression, followed by a plateau when TE sequences make up ~50% of the bp content of the non-recombining regions and the authors provide convincing evidence that TE accumulation occurs in bursts (as a result of either increased transposition or less efficient purifying selection). Remarkably, the density of TE sequences within non-recombining regions correlates positively with that of TE sequences located elsewhere in the genome, a finding that the authors take to favor the reservoir hypothesis, whereby most transposition events across the genome are contributed by active TEs located within non-recombining regions. Although this is the main conclusion presented in this manuscript, it remains unclear, based on the evidence presented, whether the opposite hypothesis of non-recombining regions serving as sinks (with most active TEs being located in recombining regions) rather than (or in addition to) reservoirs can be ruled out.

Indeed, the reservoir hypothesis is put into question by the absence of full-length Helitrons in a number of young non-recombining regions. Moreover, transposition bursts do not all appear to follow recombination-suppression events and the lack of any increase in TE load on the borders of non-recombining regions does not entirely rule out the possibility that TEs are the drivers of the loss of recombination. Specifically, several of the transposition bursts seem to be associated with two or more independent events of recombination-suppression. For instance, the recently-expanded clade of Copia retroelements that correspond to the light blue – *M. v. paradoxa* – and the purple – *M. lichnidis-dioicae* / *M. silenes-dioicae* events is difficult to explain if the bursts necessarily follow the suppression events (barring horizontal TE transfer between these distantly related species, an unlikely possibility given the analyses presented in the manuscript).

At the very least, the authors should provide better estimates of the timing of bursts in relation to recombination-suppression events. To this end, they should date more precisely the bursts based on the maximal LTR divergence, or find in the within-species phylogeny clear cases where a TE copy within a non-recombining region is at the base of a burst across the genome. In any case, a full validation of the reservoir hypothesis would require direct measurements of TE activity contributed respectively by the non-recombining and recombining compartments of the genome, an endeavor that is well beyond the scope of this study.

Overall, the manuscript is well written, pleasant to read and the analyses that are presented appear sound and informative. Despite the important limitations mentioned above and which the authors need to address properly in a revised manuscript, this is a very elegant study of the dynamics of TE accumulation in relation to recombination and as such it should be of broad interest to the evolutionary genomics community.

Other comments:

The Introduction and Discussion are overly detailed. Simplify.

L.59: increases > predicted to increase.

L.200: A brief description of the TE annotation pipeline used and of the origin of the genome sequence data is needed.

L. 255: Fig. 2D instead of 2C.

Fig. 4: The mention that the genealogy is built based on LTR sequences is missing.

L.360: The data does not suggest so far that the transposition occurs from the non-recombining region to the autosomes and not the opposite way.

L.361-362: Without invoking the reservoir hypothesis, this pattern could be explained solely by relaxed purifying selection within non-recombining regions.

L.367: LTR sequences.

L. 370: coincides with, not necessarily follows.

L. 489: Given the insertion preference of Copia/Ty1 retroelements for genes, especially incorporating the histone variant H2A.Z throughout their body (Quadrona et al. Nat Comm 2019), their accumulation in non-recombining regions is more likely to be indirect and to reflect less efficient purifying selection.

L. 497: is associated with rather than followed.

L. 498: the direction of the effect is unclear.

Reviewer #2 (Remarks to the Author):

NCOMMS-22-53309

Dynamics of transposable element accumulation in the non-recombining regions of mating-type chromosomes in anther-smut fungi

Accept with revisions

The authors conducted an analysis of the dynamics of transposable elements (TEs) in the mating-type locus of 16 species of *Microbotryum*, a fungal plant pathogen. They used a single individual per species and focused on TE accumulation in 21 regions along the mating-type locus differing on the time in which recombination ceased to occur. These 21 regions are described in previous publications, and vary according to species (e.g., not all species share the same regions). The novelty relies on the association of TEs with the timing in which recombination ceased to occur across the mating-type locus and possible consequences across the genome. By de novo TE annotation and associating TEs to various genomic compartments outside the mating-type locus, authors try to disentangle the timing of TE burst events, accumulation of TEs in non-recombining regions, and the role of TE-reservoir. The authors convincingly show the role of Copia and Ty3 elements in past TE expansion events globally affecting the genome, regardless of recombination frequency. However, when comparing traits across different species it is often advisable to account for the possibility of a phylogenetic signal influencing trait comparisons and correlations. In my comments, I suggest an additional test that would allow them to account for it.

For all comparisons among different species, as for instance in Figure 7, Figure S7, Table S4, Table S6, I would recommend taking in consideration the phylogenetic signal. One suggestion would be considering the statistical method proposed by Grafen (<https://doi.org/10.1098/rstb.1989.0106>), which can be implemented in R environment in using the function "gls" from "nlme" package.

Considering information at the population scale within a given *Microbotryum* species, would the TE dynamics observed be maintained? Also, are dates of recombination suppression similar between isolates of a same species?

Minor comments:

In Figure 1B, *M. intermedium* has no region in black (associated to non-recombining region) while in figure S2 this species has a non-recombining region size larger than zero. Is the representation in figure S2 across the entire genome?

Line 453-456: Would it be possible that a less efficient RIP contributes to the accumulation of TEs?
Would it be possible to compare the RIP measures per class of TEs?

Line 459-462: I would suggest considering other mechanisms that would act to prevent the increase in TE copy number over time in a population (e.g., TEs disrupting essential gene function or rare back mutations counteracting Muller's ratchet).

Reviewer #1 (Remarks to the Author):

This manuscript by Duhamel et al presents a comprehensive analysis of the dynamics of TE accumulation within non-recombining regions of the genome of 15 anther-smut fungal species. Specifically, the authors have exploited whole genome long-read sequencing data previously obtained for 15 *Microbotryum* species where regions of recombination suppression between mating-type loci have evolved independently at different times in the past to determine the potential impact of such recombination suppression on TE accumulation. As expected, they observe a very clear time-dependent overaccumulation of TEs within non-recombining regions compared to the rest of the genome, which is mostly contributed by Copia and Ty3 superfamilies of LTR retroelements and Helitron rolling-circle DNA transposons. More precisely, there appears to be a rapid increase in TE density within 1.5My of recombination suppression, followed by a plateau when TE sequences make up ~50% of the bp content of the non-recombining regions and the authors provide convincing evidence that TE accumulation occurs in bursts (as a result of either increased transposition or less efficient purifying selection). Remarkably, the density of TE sequences within non-recombining regions correlates positively with that of TE sequences located elsewhere in the genome, a finding that the authors take to favor the reservoir hypothesis, whereby most transposition events across the genome are contributed by active TEs located within non-recombining regions.

>>We thank the referee very much for these positive comments.

***Although this is the main conclusion presented in this manuscript, it remains unclear, based on the evidence presented, whether the opposite hypothesis of non-recombining regions serving as sinks (with most active TEs being located in recombining regions) rather than (or in addition to) reservoirs can be ruled out.**

>>The reservoir hypothesis was not meant actually to be the main conclusion of our manuscript, which was instead the tempo of transposable element accumulation, as highlighted in the title. We have tried to make this point clearer.

We have added new analyses, in particular a comparison of the age of copies in autosomes compared to non-recombining regions, showing that TE copies are older in non-recombining regions, in agreement with the reservoir hypothesis, although, again, not constituting definitive proof.

We agree that our data does not completely rule out the opposite of the reservoir hypothesis and we have tried to make this clearer too : we have deleted statements in the abstract, the end of the introduction, the results, and we discuss the alternative hypothesis more thoroughly.

However, we found a clear and positive relationship between TE content in non-recombining regions and the age of recombination suppression. Such a pattern would not be predicted by the opposite of the reservoir hypothesis ; or, this would mean that, by chance, the species with older non-recombining regions have accumulated more TEs in autosomes than the species with younger non-recombining regions, for another (unknown) reason than the recombination suppression. While not formally ruled out by our data, this sounds highly unlikely and would be an extraordinary happy coincidence. We have tried to make this clearer too.

***Indeed, the reservoir hypothesis is put into question by the absence of full-length Helitrons in a number of young non-recombining regions.**

>>We do not see any contradiction here? The young non-recombining regions have precisely accumulated little TE load and may not have acquired active Helitrons yet. They may acquire them later by jumps from older non-recombining regions. Note however on Supplementary Figure S19 that some young strata do have intact Helitrons.

***Moreover, transposition bursts do not all appear to follow recombination-suppression events**

>>See our answer below on this point.

and the lack of any increase in TE load on the borders of nonrecombining regions does not entirely rule out the possibility that TEs are the drivers of the loss of recombination.

>>We agree and this is discussed in the discussion P21, we have deleted the earlier statements that could let readers think we concluded the opposite.

***Specifically, several of the transposition bursts seem to be associated with two or more independent events of recombination-suppression. For instance, the recently-expanded clade of Copia retroelements that correspond to the light blue – *M. v. paradoxa* – and the purple – *M. lichnidis-dioicae* / *M. silenes-dioicae* events is difficult to explain if the bursts necessarily follow the suppression events (barring horizontal TE transfer between these distantly related species, an unlikely possibility given the analyses presented in the manuscript).**

>> We now discuss the clade with mixed blue and purple colors at the bottom right of the figure. This is the only exception to the pattern of bursts being restricted to recombination suppression events and *M.v. paradoxa* is precisely the species in which we detected introgression from a distant species in the non-recombining region in a previous study (Carpentier et al. 2022). We thank the referee for pointing to this interesting observation.

***At the very least, the authors should provide better estimates of the timing of bursts in relation to recombination-suppression events. To this end, they should date more precisely the bursts based on the maximal LTR divergence, or find in the within-species phylogeny clear cases where a TE copy within a non-recombining region is at the base of a burst across the genome.**

>>Attempts to dating bursts based on LTR sequences with absolute time lead to unreliable estimates given the small length of the LTRs and the huge confidence intervals

of the available mutation rate estimates. We have nevertheless tried an absolute datation using the same method as in Gupta et al., 2023 (<https://www.nature.com/articles/s41467-023-37551-4>); as expected, confidence intervals are larger than point estimates (see Table at the end of this letter). We therefore think that such estimates are minimally informative, or even misleading. The relative dating in the genealogy in Figure 6 is much more useful and robust, and the only exception highlighted above can be explained by introgressions previously reported. Finding a TE copy at the basis of a burst across the genome cannot reliably settle the question either, as many copies are missing in the genealogy, having been lost or with a LTR missing or not identified, so we likely lack the original mother copy of a burst.

***In any case, a full validation of the reservoir hypothesis would require direct measurements of TE activity contributed respectively by the non-recombining and recombining compartments of the genome, an endeavor that is well beyond the scope of this study.**

>>We agree and this has been added Lines 455-456.

***Overall, the manuscript is well written, pleasant to read and the analyses that are presented appear sound and informative. Despite the important limitations mentioned above and which the authors need to address properly in a revised manuscript, this is a very elegant study of the dynamics of TE accumulation in relation to recombination and as such it should be of broad interest to the evolutionary genomics community.**

>>We thank the referee again for this positive assessment and the constructive comments above.

***Other comments:**

***The Introduction and Discussion are overly detailed. Simplify.**

>>We have shortened and simplified the introduction and discussion to remain below the 6,000 word limit of the journal. We have also shortened the abstract to meet the 150 word requirement.

***L.59: increases > predicted to increase.**

>>This has been corrected

***L.200: A brief description of the TE annotation pipeline used and of the origin of the genome sequence data is needed.**

>>This has been added.

***L. 255: Fig. 2D instead of 2C.**

>>This has been corrected

***Fig. 4: The mention that the genealogy is built based on LTR sequences is missing.**

>>This has been added in the legend.

***L.360: The data does not suggest so far that the transposition occurs from the non-recombining region to the autosomes and not the opposite way.**

>>This has been deleted

***L.361-362: Without invoking the reservoir hypothesis, this pattern could be explained solely by relaxed purifying selection within non-recombining regions.**

>>This has been deleted (although we do not see how the coincidence of TE accumulation in both autosomes and non-recombining regions could be due to relaxed purifying selection within non-recombining regions alone otherwise, but we discuss this more extensively in the discussion now).

***L.367: LTR sequences.**

>>This has been added

***L. 370: coincides with, not necessarily follows.**

>> This has been changed as suggested.

***L. 489: Given the insertion preference of Copia/Ty1 retroelements for genes, especially incorporating the histone variant H2A.Z throughout their body (Quadrana et al. Nat Comm 2019), their accumulation in non-recombining regions is more likely to be indirect and to reflect less efficient purifying selection.**

>>All TE types accumulate in non-recombining regions because of less efficient purifying selection. We have added Lines 424-429 that the purifying selection on autosomes is likely stronger for these elements due to preferential insertions in genes, which could indeed make the differential of selection higher between recombining and non-recombining regions.

***L. 497: is associated with rather than followed.**

>>This has been changed as suggested.

***L. 498: the direction of the effect is unclear.**

>>We have tried to tone down the claim and to discuss it further : « The direction of the effect is unlikely to be in the other direction, i.e., TE bursts in the autosomes that would invade non-recombining regions precisely when recombination suppression evolves, or this would imply that TE activity in autosomes positively correlated to the age of the non-recombining regions for a different reason than recombination suppression ».

***Reviewer #2 (Remarks to the Author): NCOMMS-22-53309 Dynamics of transposable element accumulation in the non-recombining regions of mating-type chromosomes in anther-smut fungi** Accept with revisions The authors conducted an analysis of the dynamics of transposable elements (TEs) in the mating-type locus of 16 species of *Microbotryum*, a fungal plant pathogen. They used a single individual per species and focused on TE accumulation in 21 regions along the mating-type locus differing on the time in which recombination ceased to occur. These 21 regions are described in

previous publications, and vary according to species (e.g., not all species share the same regions). The novelty relies on the association of TEs with the timing in which recombination ceased to occur across the mating-type locus and possible consequences across the genome. By de novo TE annotation and associating TEs to various genomic compartments outside the mating-type locus, authors try to disentangle the timing of TE burst events, accumulation of TEs in non-recombining regions, and the role of TE-reservoir. The authors convincingly show the role of Copia and Ty3 elements in past TE expansion events globally affecting the genome, regardless of recombination frequency.

>>We thank the referee very much for these positive comments.

***However, when comparing traits across different species it is often advisable to account for the possibility of a phylogenetic signal influencing trait comparisons and correlations. In my comments, I suggest an additional test that would allow them to account for it.**

For all comparisons among different species, as for instance in Figure 7, Figure S7, Table S4, Table S6, I would recommend taking in consideration the phylogenetic signal. One suggestion would be considering the statistical method proposed by Grafen (<https://doi.org/10.1098/rstb.1989.0106>), which can be implemented in R environment in using the function "gls" from "nlme" package.

>>We have implemented the recommended approach and no phylogenetic signal was detected. Indeed, we had been careful to study independent recombination suppression events : for the strata shared by multiple species, we took the mean values across the species derived from the same recombination suppression event.

***Considering information at the population scale within a given Microbotryum species, would the TE dynamics observed be maintained? Also, are dates of recombination suppression similar between isolates of a same species?**

>> We have added in the manuscript analyses on an additional genome of *M. lychnidis-dioicae* (sequence data and genome assemblies have been submitted to Genbank, we are waiting for the project ID that will be added upon manuscript acceptance), which shows that i) it has similar dS levels as the reference genome, ii) TE copies of the two *M. lychnidis-dioicae* strains are intermingled in each compartment, suggesting that the pattern are representative of the species and not strain-specific.

We do not have other genomes so far with sufficient assembly quality to quantify the variation of TE content within species in non-recombining regions: the non-recombining regions are too fragmented for reliably estimating TE content in our current resequencing (Illumina) datasets. We agree that this will be interesting to assess in future studies. The dates of recombination suppression events are much older than speciation events (Branco et al. 2018 ; Duhamel et al. 2022), which suggests that all strains within species will show similar dates of recombination suppression events. The TE dynamics within species will be very interesting to study, but goes much beyond the present study and will require multiple high-quality genome assemblies per species, and will address different questions and timescales than the present study.

***Minor comments: In Figure 1B, *M. intermedium* has no region in black (associated to non-recombining region) while in figure S2 this species has a non-recombining region size larger than zero. Is the representation in figure S2 across the entire genome?**

>> Thanks for noting this mistake, we have corrected the supplementary figure S2.

***Line 453-456: Would it be possible that a less efficient RIP contributes to the accumulation of TEs? Would it be possible to compare the RIP measures per class of TEs?**

>> Our data suggest that RIP is not less efficient in non-recombining regions : Lines 215-216 «The TEs had more RIP-like signatures in non-recombining regions than in autosomes and pseudo-autosomal regions in most species ». Lines 370-371: « suggesting that a process similar to RIP efficiently mutates TE copies in young non-

recombining regions ». We have specified this in the discussion Lines 413-414. However, the referee is right that the TE classes expanded in non-recombining regions are less RIPed than other TE classes (although this effect is lower in the non-recombining region, i.e. these TEs are even less RIPed in autosomes), we have added this analysis and we thank the referee for this interesting insight.

Line 459-462: I would suggest considering other mechanisms that would act to prevent the increase in TE copy number over time in a population (e.g., TEs disrupting essential gene function or rare back mutations counteracting Muller's ratchet).

>>We have added a sentence Lines 410-411 : « Strong selection against TE copies disrupting essential gene function or rare back mutations may slow down Muller's ratchet but cannot stop it ».

>>Attempt of dating bursts using LTR divergence: estimate date (in years) for each burst and standard deviation

Burst	date estimate	standard deviation
allCopia_LTRseq_LTR1_MsaSac__burst_1	1,02E+07	8,50E+07
allCopia_LTRseq_LTR1_MsaSac__burst_2	9,95E+06	2,33E+07
allCopia_LTRseq_LTR1_MsaSac__burst_5	8,42E+06	7,10E+07
allCopia_LTRseq_LTR1_MsaSac__burst_6	1,18E+07	9,09E+07
allCopia_LTRseq_LTR1_MsaSac__burst_7	1,07E+07	4,43E+07
allCopia_LTRseq_LTR1_MsaSac__burst_8	1,30E+07	8,84E+07
allCopia_LTRseq_LTR1_MsdSdi__burst_10	3,20E+06	7,72E+07
allCopia_LTRseq_LTR1_MsdSdi__burst_11	1,21E+07	5,29E+07

allCopia_LTRseq_LTR1_MsdSdi__burst_1	2,10E+07	8,43E+07
allCopia_LTRseq_LTR1_MsdSdi__burst_2	8,89E+06	8,77E+07
allCopia_LTRseq_LTR1_MsdSdi__burst_3	8,37E+06	4,04E+07
allCopia_LTRseq_LTR1_MsdSdi__burst_4	1,17E+07	4,86E+07
allCopia_LTRseq_LTR1_MsdSdi__burst_5	6,55E+06	6,79E+07
allCopia_LTRseq_LTR1_MsdSdi__burst_6	9,13E+06	7,84E+07
allCopia_LTRseq_LTR1_MsdSdi__burst_7	2,63E+05	9,36E+07
allCopia_LTRseq_LTR1_MsdSdi__burst_9	7,71E+06	5,34E+07
allCopia_LTRseq_LTR1_MspSpr__burst_1	4,83E+06	8,62E+07
allCopia_LTRseq_LTR1_MspSpr__burst_2	1,38E+07	6,62E+07
allCopia_LTRseq_LTR1_MspSpr__burst_3	9,73E+06	4,84E+07
allCopia_LTRseq_LTR1_MspSpr__burst_4	1,79E+07	9,35E+07
allCopia_LTRseq_LTR1_MspSpr__burst_5	1,52E+07	7,69E+07
allCopia_LTRseq_LTR1_MspSpr__burst_6	2,98E+07	5,17E+07
allCopia_LTRseq_LTR1_MspSra__burst_10	4,91E+06	7,84E+07
allCopia_LTRseq_LTR1_MspSra__burst_11	1,23E+07	5,41E+07
allCopia_LTRseq_LTR1_MspSra__burst_1	1,38E+07	7,08E+07
allCopia_LTRseq_LTR1_MspSra__burst_12	1,18E+07	2,43E+07
allCopia_LTRseq_LTR1_MspSra__burst_2	7,88E+06	9,05E+07

allCopia_LTRseq_LTR1_MspSra__burst_3	1,03E+07	4,32E+07
allCopia_LTRseq_LTR1_MspSra__burst_4	6,07E+06	6,20E+07
allCopia_LTRseq_LTR1_MspSra__burst_5	5,08E+06	8,46E+07
allCopia_LTRseq_LTR1_MspSra__burst_6	8,10E+06	1,98E+07
allCopia_LTRseq_LTR1_MspSra__burst_7	2,98E+06	5,31E+07
allCopia_LTRseq_LTR1_MspSra__burst_8	9,78E+06	7,81E+07
allCopia_LTRseq_LTR1_MspSra__burst_9	2,13E+06	9,36E+07
allCopia_LTRseq_LTR1_MspSsc__burst_10	5,61E+06	8,45E+07
allCopia_LTRseq_LTR1_MspSsc__burst_11	1,54E+07	1,80E+07
allCopia_LTRseq_LTR1_MspSsc__burst_12	1,32E+07	4,22E+07
allCopia_LTRseq_LTR1_MspSsc__burst_13	6,62E+06	6,18E+07
allCopia_LTRseq_LTR1_MspSsc__burst_1	4,80E+06	7,92E+07
allCopia_LTRseq_LTR1_MspSsc__burst_2	8,50E+06	5,83E+07
allCopia_LTRseq_LTR1_MspSsc__burst_3	5,78E+06	3,65E+07
allCopia_LTRseq_LTR1_MspSsc__burst_4	6,32E+06	8,54E+07
allCopia_LTRseq_LTR1_MspSsc__burst_5	4,33E+06	6,41E+07
allCopia_LTRseq_LTR1_MspSsc__burst_6	3,33E+06	4,70E+07
allCopia_LTRseq_LTR1_MspSsc__burst_7	8,43E+06	9,22E+07
allCopia_LTRseq_LTR1_MspSsc__burst_9	3,22E+06	5,10E+07

allCopia_LTRseq_LTR1_MviLyf__burst_1	5,24E+06	7,28E+07
allCopia_LTRseq_LTR1_MviLyf__burst_2	1,03E+07	6,40E+07
allCopia_LTRseq_LTR1_MviLyf__burst_3	4,13E+06	4,58E+07
allCopia_LTRseq_LTR1_MviLyf__burst_4	1,10E+07	7,95E+07
allCopia_LTRseq_LTR1_MviLyf__burst_5	4,25E+06	5,97E+07
allCopia_LTRseq_LTR1_MviSco__burst_10	9,35E+06	8,52E+07
allCopia_LTRseq_LTR1_MviSco__burst_1	1,07E+07	6,71E+07
allCopia_LTRseq_LTR1_MviSco__burst_2	6,42E+06	6,16E+07
allCopia_LTRseq_LTR1_MviSco__burst_3	1,34E+07	4,02E+07
allCopia_LTRseq_LTR1_MviSco__burst_4	8,42E+06	9,64E+07
allCopia_LTRseq_LTR1_MviSco__burst_5	8,18E+06	7,97E+07
allCopia_LTRseq_LTR1_MviSco__burst_6	6,16E+06	7,47E+07
allCopia_LTRseq_LTR1_MviSco__burst_7	9,85E+06	5,09E+07
allCopia_LTRseq_LTR1_MviSco__burst_8	1,08E+07	4,69E+07
allCopia_LTRseq_LTR1_MviSco__burst_9	8,21E+06	9,21E+07
allCopia_LTRseq_LTR1_MviSic__burst_1	1,12E+07	9,58E+07
allCopia_LTRseq_LTR1_MviSic__burst_2	6,90E+06	7,94E+07
allCopia_LTRseq_LTR1_MviSic__burst_3	1,45E+07	5,90E+07
allCopia_LTRseq_LTR1_MviSic__burst_4	1,83E+07	6,35E+07

allCopia_LTRseq_LTR1_MviSic__burst_6	1,24E+07	9,16E+07
allCopia_LTRseq_LTR1_MviSic__burst_7	4,24E+06	7,26E+07
allCopia_LTRseq_LTR1_MviSic__burst_8	1,07E+07	9,31E+07
allCopia_LTRseq_LTR1_MviSpa__burst_10	1,93E+07	6,29E+07
allCopia_LTRseq_LTR1_MviSpa__burst_11	1,17E+07	4,53E+07
allCopia_LTRseq_LTR1_MviSpa__burst_1	1,53E+07	9,62E+07
allCopia_LTRseq_LTR1_MviSpa__burst_12	1,06E+06	9,12E+07
allCopia_LTRseq_LTR1_MviSpa__burst_13	8,13E+06	7,25E+07
allCopia_LTRseq_LTR1_MviSpa__burst_2	8,29E+06	5,02E+07
allCopia_LTRseq_LTR1_MviSpa__burst_3	1,07E+07	7,46E+07
allCopia_LTRseq_LTR1_MviSpa__burst_4	9,86E+06	6,68E+07
allCopia_LTRseq_LTR1_MviSpa__burst_5	2,23E+07	8,77E+07
allCopia_LTRseq_LTR1_MviSpa__burst_6	1,84E+07	3,83E+07
allCopia_LTRseq_LTR1_MviSpa__burst_7	7,88E+06	6,10E+07
allCopia_LTRseq_LTR1_MviSpa__burst_8	2,01E+07	3,58E+07
allCopia_LTRseq_LTR1_MviSpa__burst_9	3,06E+07	5,81E+07
allCopia_LTRseq_LTR1_MvSI__burst_1	7,38E+06	8,69E+07
allCopia_LTRseq_LTR1_MvSI__burst_2	9,33E+06	3,74E+07
allCopia_LTRseq_LTR1_MvSI__burst_3	6,74E+06	6,05E+07

allCopia_LTRseq_LTR1_MvSI__burst_4	1,60E+07	7,96E+07
allCopia_LTRseq_LTR1_MvSI__burst_5	9,48E+06	9,60E+07
allCopia_LTRseq_LTR1_MvSI__burst_6	9,60E+06	4,99E+07
allCopia_LTRseq_LTR1_MvSI__burst_7	1,13E+07	7,41E+07
allCopia_LTRseq_LTR1_MvSI__burst_8	1,70E+06	9,19E+07
allCopia_LTRseq_LTR1_MvSI__burst_9	7,49E+06	4,66E+07
allGypsy_LTRseq_LTR1_MsdSdi__burst_2	5,79E+06	4,76E+07
allGypsy_LTRseq_LTR1_MsdSdi__burst_3	1,28E+07	6,51E+07
allGypsy_LTRseq_LTR1_MsdSdi__burst_4	1,76E+07	7,57E+07
allGypsy_LTRseq_LTR1_MsdSdi__burst_5	7,17E+06	9,25E+07
allGypsy_LTRseq_LTR1_MsdSdi__burst_6	1,11E+07	6,55E+06
allGypsy_LTRseq_LTR1_MviSpa__burst_1	6,39E+06	5,48E+07
allGypsy_LTRseq_LTR1_MviSpa__burst_2	1,43E+07	7,91E+07
allGypsy_LTRseq_LTR1_MviSpa__burst_3	1,88E+07	9,42E+07
allGypsy_LTRseq_LTR1_MviSpa__burst_4	1,28E+07	9,19E+07

REVIEWERS' COMMENTS

Reviewer #1 (Remarks to the Author):

The authors have answered comprehensively and satisfactorily all of the points raised in both reviews.

Reviewer #2 (Remarks to the Author):

The authors have elegantly addressed all my concerns in the revision.

Reviewer #3 (Remarks to the Author):

In this revised version of their manuscript, Duhamel et al. have satisfactorily addressed all points raised during the first round of reviews. Notably, their re-wording throughout the text now takes into account that the evidence provided in this study is very much in agreement with non-recombining regions acting as a reservoir but cannot be taken as a direct demonstration. After a minor edit (see below), this study is ready for publication.

L310-311: the current wording implies that TEs being less often purged by selection in non-recombining regions is in agreement with the reservoir hypothesis. However, you can have TEs being less often purged without any reservoir effect. To clarify, I would suggest rephrasing this sentence into “LTR divergence indicated that [...], which is expected given that they are less often purged by selection in non-recombining regions and in agreement with the reservoir hypothesis”.

Reviewer #3 (Remarks to the Author):

In this revised version of their manuscript, Duhamel et al. have satisfactorily addressed all points raised during the first round of reviews. Notably, their re-wording throughout the text now takes into account that the evidence provided in this study is very much in agreement with non-recombining regions acting as a reservoir but cannot be taken as a direct demonstration. After a minor edit (see below), this study is ready for publication.

L310-311: the current wording implies that TEs being less often purged by selection in non-recombining regions is in agreement with the reservoir hypothesis. However, you can have TEs being less often purged without any reservoir effect. To clarify, I would suggest rephrasing this sentence into “LTR divergence indicated that [...], which is expected given that they are less often purged by selection in non-recombining regions and in agreement with the reservoir hypothesis”.

>>We have made the requested change.